# PUMILIO hyperactivity drives premature aging of *Norad*-deficient mice

**Florian Kopp[1], Mahmoud M Elguindy[1†], Mehmet E Yalvac[2,3†], He Zhang[4,5], Beibei Chen[4,5], Frank A Gillett[1], Sungyul Lee[1], Sushama Sivakumar[6], Hongtao Yu[6,7], Yang Xie[4,5,8], Prashant Mishra[9], Zarife Sahenk[2,10,11], Joshua T Mendell[1,7,8,12]***

[1]Department of Molecular Biology, University of Texas Southwestern Medical Center, Dallas, United States; [2]Center for Gene Therapy, Nationwide Children's Hospital, Columbus, United States; [3]Department of Neurology, The Ohio State University, Columbus, United States; [4]Quantitative Biomedical Research Center, University of Texas Southwestern Medical Center, Dallas, United States; [5]Department of Clinical Sciences, University of Texas Southwestern Medical Center, Dallas, United States; [6]Department of Pharmacology, University of Texas Southwestern Medical Center, Dallas, United States; [7]Howard Hughes Medical Institute, University of Texas Southwestern Medical Center, Dallas, United States; [8]Harold C Simmons Comprehensive Cancer Center, University of Texas Southwestern Medical Center, Dallas, United States; [9]Children's Medical Center Research Institute, University of Texas Southwestern Medical Center, Dallas, United States; [10]Department of Pediatrics, The Ohio State University, Columbus, United States; [11]Department of Neurology, The Ohio State University, Columbus, United States; [12]Hamon Center for Regenerative Science and Medicine, University of Texas Southwestern Medical Center, Dallas, United States

**\*For correspondence:**
joshua.mendell@utsouthwestern.edu

[†]These authors contributed equally to this work

**Competing interests:** The authors declare that no competing interests exist.

**Abstract** Although numerous long noncoding RNAs (lncRNAs) have been identified, our understanding of their roles in mammalian physiology remains limited. Here, we investigated the physiologic function of the conserved lncRNA *Norad* in vivo. Deletion of *Norad* in mice results in genomic instability and mitochondrial dysfunction, leading to a dramatic multi-system degenerative phenotype resembling premature aging. Loss of tissue homeostasis in *Norad*-deficient animals is attributable to augmented activity of PUMILIO proteins, which act as post-transcriptional repressors of target mRNAs to which they bind. *Norad* is the preferred RNA target of PUMILIO2 (PUM2) in mouse tissues and, upon loss of *Norad*, PUM2 hyperactively represses key genes required for mitosis and mitochondrial function. Accordingly, enforced *Pum2* expression fully phenocopies *Norad* deletion, resulting in rapid-onset aging-associated phenotypes. These findings provide new insights and open new lines of investigation into the roles of noncoding RNAs and RNA binding proteins in normal physiology and aging.
DOI: https://doi.org/10.7554/eLife.42650.001

## Introduction

Long noncoding RNAs (lncRNAs) comprise a heterogeneous class of transcripts that are defined by a sequence length greater than 200 nucleotides and the lack of a translated open-reading frame (ORF). lncRNAs have been proposed to perform a variety of cellular functions including regulation of gene expression in *cis* and *trans*, modulation of functions of RNAs and proteins to which they bind, and organization of nuclear architecture (*Batista and Chang, 2013*). Although they have been

**eLife digest** Only a tiny portion of our genetic material contains the information required to create proteins, the workhorses of the body. The rest of our DNA, however, is not useless: some of it can be transcribed to create molecules known as non-coding RNAs, which are increasingly scrutinized by scientists.

For example, a non-coding RNA called *NORAD* acts as a guardian of the genome by reducing the activity of a protein named PUMILIO. Without *NORAD*, PUMILIO becomes overactive, and this causes problems as genetic information is split between two 'daughter cells' when a cell divides.

Defects in the amount of genetic material in cells have been linked with faster aging in animals. In addition, some studies suggest that as animals get older, the levels of *NORAD* in the body decrease, while the levels of PUMILIO increase. However, the precise role that *NORAD* may play in aging remains unclear.

To address this question, Kopp et al. engineered mutant mice that lack *Norad* (the mouse equivalent of human *NORAD*) and carefully monitored how they grew and developed. The animals looked normal at birth, but they seemed to age faster: for instance, their fur became thin and gray, and their brains developed age-related abnormalities much sooner than normal mice.

At the level of individual cells, losing *Norad* was also associated with problems often seen in old age. The mutant animals were more likely to have incorrect amounts of genetic information in their cells, and they had defects in the cell compartments that create the energy necessary for life. Further experiments showed that these issues were driven by PUMILIO being hyperactive. Overall, the work by Kopp et al. reveal that the non-coding RNA *Norad* is essential to keep PUMILIO activity in check and to prevent problems associated with aging from appearing in young animals. Further studies are now needed to take a closer look at how *NORAD* and other non-coding RNAs keep us healthy.

DOI: https://doi.org/10.7554/eLife.42650.002

estimated to number in the tens of thousands (*Harrow et al., 2012*), the biological significance of the vast majority of lncRNAs remains to be established. This is due, in part, to the generally low abundance and poor evolutionary conservation of most lncRNAs, which has limited our ability to interrogate their biochemical functions as well as their biologic roles in vivo using model organisms. Moreover, while genetic studies in mice have uncovered important functions for selected mammalian lncRNA loci in development and disease states (*Arun et al., 2016*; *Sauvageau et al., 2013*), it has often been challenging to connect specific lncRNA-driven phenotypes to defined RNA-mediated functions. As a result, our broad understanding of how the molecular pathways controlled by lncRNAs impact development and physiology remains limited.

*Noncoding RNA activated by DNA damage* (*NORAD*) is a recently described lncRNA that is distinguished from the majority of transcripts in this class due to its high abundance in mammalian cells and strong evolutionary conservation across mammalian species (*Lee et al., 2016*; *Tichon et al., 2016*). Studies in human cells have established that this RNA functions as a strong negative regulator of PUMILIO1 (PUM1) and PUMILIO2 (PUM2), RNA binding proteins (RBPs) that belong to the deeply conserved family of Pumilio and Fem3 binding factor (PUF) proteins. PUM1/2 bind specifically to the eight nucleotide (nt) PUMILIO response element (PRE) (UGUANAUA), which is often located in the 3' UTR of mRNAs. Binding of PUM1/2 to these sites triggers accelerated deadenylation, reduced translation, and turnover of mRNA targets (*Miller and Olivas, 2011*; *Quenault et al., 2011*). With the capacity to bind a large fraction of PUM1/2 within the cell, *NORAD* limits the availability of these proteins to repress target mRNAs (*Lee et al., 2016*; *Tichon et al., 2016*). Consequently, inactivation of *NORAD* results in PUMILIO hyperactivity with augmented repression of a program of target mRNAs that includes key regulators of mitosis, DNA repair, and DNA replication. Dysregulation of these genes results in dramatic genomic instability in *NORAD*-deficient cells (*Lee et al., 2016*). In accordance with this model, PUM2 overexpression is sufficient to phenocopy, while PUM1/2 loss-of-function is sufficient to suppress, the *NORAD* knockout phenotype in human cells. Recent work has identified additional RNA-binding proteins that interact with *NORAD* including SAM68, which facilitates PUMILIO antagonism by this lncRNA (*Tichon et al., 2018*), and RBMX, an RNA binding protein

that contributes to the DNA damage response (*Munschauer et al., 2018*). While it has not yet been demonstrated that *NORAD*:RBMX interaction is essential to maintain genomic stability, this intriguing interaction raises the possibility of additional functions for *NORAD* beyond regulation of PUMILIO activity.

Although PUF proteins are deeply conserved across eukaryotic species, the emergence of *NORAD* specifically within mammals suggests the existence of strong selective pressure to maintain tight control of PUMILIO activity within this lineage. In mice, PUM1 and PUM2 loss-of-function has been linked to behavioral abnormalities, elevated neuronal excitability, and impaired neurogenesis, while inactivation of PUM1 reduces fertility in males and females (*Goldstrohm et al., 2018*). Interestingly, mammalian neurons are exquisitely sensitive to reduced dosage of PUMILIO, with only a 25% to 50% reduction in PUM1 expression resulting in neurodegeneration in both human and mouse brain (*Gennarino et al., 2018*; *Gennarino et al., 2015*). The existence of *NORAD* suggests that hyperactivity of PUMILIO, expected to occur in the absence of this lncRNA, may also result in deleterious effects. While studies in cell lines have demonstrated that *NORAD* loss or PUMILIO overexpression results in genomic instability, the consequences of mammalian PUMILIO hyperactivity in vivo have yet to be examined.

Here, we report an investigation of the physiologic role of the *Norad*-PUMILIO axis through the generation and characterization of *Norad*-deficient and *Pum2* transgenic mouse lines. While deletion of *Norad* does not overtly impact development, mice lacking this lncRNA develop a dramatic multi-system degenerative phenotype that resembles premature aging. Loss of *Norad* results in PUMILIO hyperactivity and repression of genes that are essential for normal mitosis, leading to the accumulation of aneuploid cells in *Norad*-deficient tissues. Unexpectedly, *Norad* loss also causes striking mitochondrial dysfunction, associated with repression of PUMILIO targets that regulate mitochondrial homeostasis. Importantly, transgenic expression of *Pum2* is sufficient to fully phenocopy *Norad* loss of function, triggering genomic instability, mitochondrial dysfunction, and rapidly-advancing aging-associated phenotypes. These findings demonstrate the importance of *Norad* in maintaining tight control of PUMILIO activity in vivo and establish a critical role for this lncRNA-RBP regulatory interaction in mammalian physiology.

## Results

### Deletion of the mouse *Norad* ortholog

A mouse ortholog of *NORAD* (*2900097C17Rik* or *Norad*), exhibiting 61% nucleotide identity with its human counterpart, is clearly identifiable on mouse chromosome 2 (*Figure 1A*). Like the human transcript, mouse *Norad* shows minimal protein-coding potential as assessed by PhyloCSF, a metric that discriminates between coding and noncoding sequences based on their evolutionary signatures (*Lin et al., 2011*) (*Figure 1—figure supplement 1A*). Both human and mouse *NORAD*/*Norad* are ubiquitously expressed throughout the body, with highest abundance in brain (*Figure 1B* and *Figure 1—figure supplement 1B*). RNA fluorescent in situ hybridization (RNA FISH) in mouse embryonic fibroblasts (MEFs), as well as fractionation experiments in a panel of mouse cell lines, revealed a predominately cytoplasmic localization of *Norad* (*Figure 1C* and *Figure 1—figure supplement 1C*), mirroring the localization of the human transcript (*Lee et al., 2016*; *Tichon et al., 2016*).

To investigate the function of the mouse *Norad* ortholog, clustered regularly interspaced short palindromic repeats (CRISPR)/Cas9-mediated genome editing was employed to delete the lncRNA-encoding sequence from the mouse genome, yielding three independent knockout lines (*Figure 1D–E* and *Figure 1—figure supplement 2A*). Importantly, expression of the neighboring genes *Epb41l1* and *Cnbd2* was unaffected in *Norad*⁻/⁻ brain and spleen (*Figure 1—figure supplement 2B*), demonstrating that deletion of the *Norad* locus does not perturb neighboring gene expression.

### *Norad* loss-of-function results in a degenerative phenotype resembling premature aging

*Norad*⁻/⁻ mice were viable, fertile, and born at the expected Mendelian frequency (*Figure 2—figure supplement 1A*). Early in life, *Norad*-deficient mice were indistinguishable from wild-type littermates, but by 6 months of age, the onset of a multi-system degenerative phenotype resembling

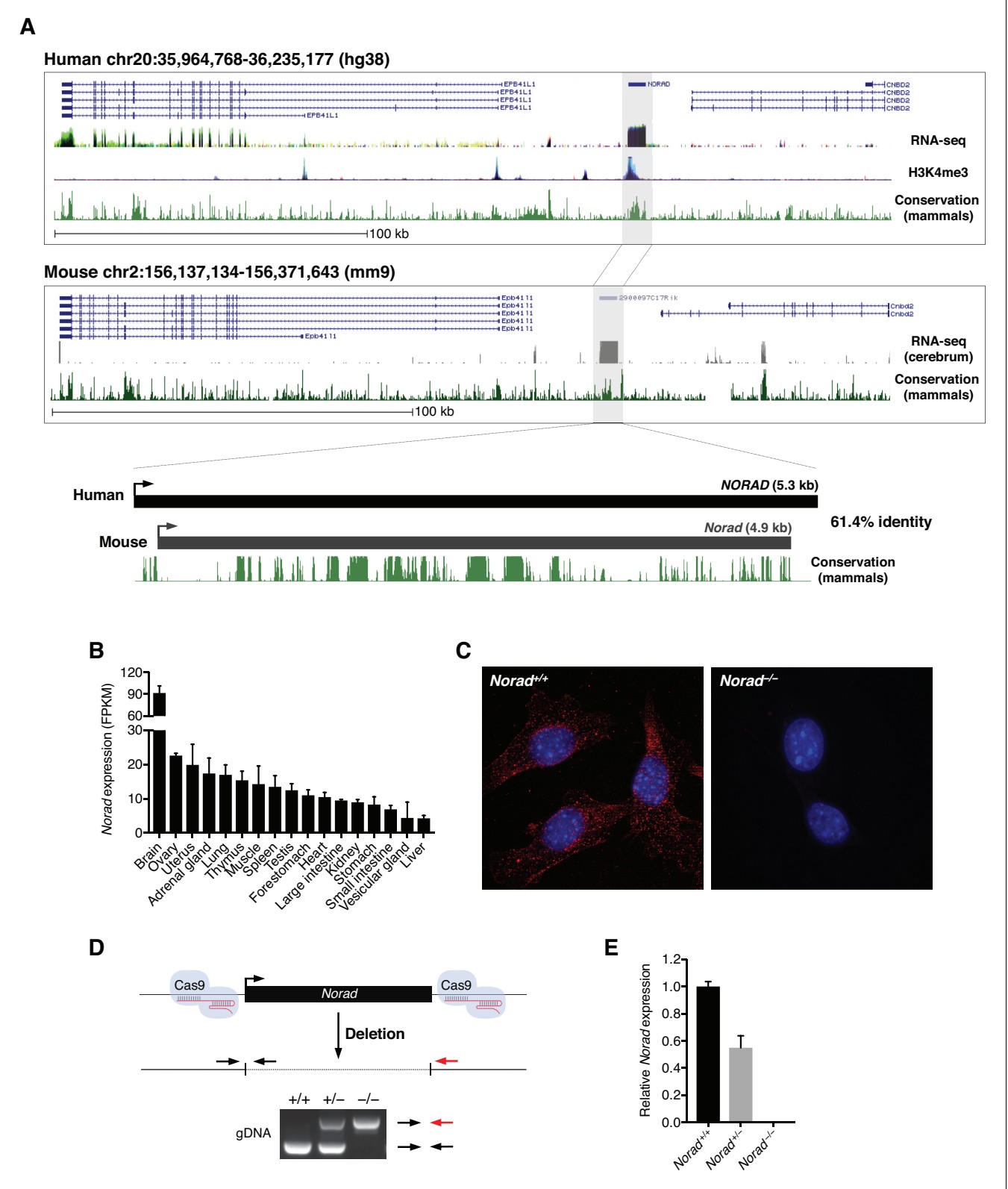

**Figure 1.** Deletion of the mouse *Norad* ortholog. (**A**) Syntenic regions of human and mouse chromosomes 20 and 2, respectively, harboring *Norad* orthologs. The following UCSC Genome Browser tracks are shown: human and mouse Refseq genes, human ENCODE regulation track with overlay of RNA-seq data from 9 cell lines and H3K4me3 data from 7 cell lines, mammalian conservation for both human and mouse (Phastcons), and mouse ENCODE UW RNA-seq data from mouse cerebrum. (**B**) *Norad* expression across 17 mouse tissues based on RNA-seq data from the Mouse

*Figure 1 continued on next page*

*Figure 1 continued*

Transcriptomic BodyMap (*Li et al., 2017*). (C) RNA FISH in immortalized MEFs of the indicated genotypes demonstrates predominantly cytoplasmic localization of *Norad*. (D) Schematic of CRISPR/Cas9-mediated genome-editing strategy used to delete *Norad* in mice. Genotyping PCR strategy and a representative genotyping result are shown. (E) *Norad* expression in primary MEF lines of the indicated genotypes, determined by qRT-PCR (n = 3 biological replicates). Data are represented as mean ± SD in (B) and (E).

DOI: https://doi.org/10.7554/eLife.42650.003

The following figure supplements are available for figure 1:

**Figure supplement 1.** Deletion of the mouse *Norad* ortholog.

DOI: https://doi.org/10.7554/eLife.42650.004

**Figure supplement 2.** Deletion of the mouse *Norad* ortholog.

DOI: https://doi.org/10.7554/eLife.42650.005

premature aging became apparent, with approximately 50% of the mice developing severe manifestations by 1 year of age (*Figure 2A–B*). This phenotype was characterized by accelerated alopecia and graying of fur in male *Norad*$^{-/-}$ mice (*Figure 2A* and *Figure 2—figure supplement 1B*), while both male and female *Norad*$^{-/-}$ mice displayed pronounced kyphosis (*Figure 2B* and *Figure 2—figure supplement 1C*). Increased kyphosis was also evident in *Norad*$^{+/-}$ animals, indicating a dose-dependent effect of *Norad* loss of function. Although body weight was comparable between cohorts of *Norad*$^{+/+}$, *Norad*$^{+/-}$, and *Norad*$^{-/-}$ mice at 1 year of age (*Figure 2—figure supplement 1D*), mice that developed outward features of aging such as kyphosis also exhibited significant weight loss accompanied by loss of total body fat and subcutaneous fat (*Figure 2C–E*). Abnormalities were also apparent in *Norad*-deficient skeletal muscle, with marked switching of fast twitch glycolytic (FTG) fibers to slow twitch oxidative (STO) fibers (*Figure 2F*), a phenomenon associated with normal muscle aging (*Ciciliot et al., 2013*). In addition, *Norad*$^{-/-}$ mice showed accelerated onset of aging-associated pathologies within the central nervous system (CNS) (*Gray and Woulfe, 2005*; *Samorajski et al., 1968*; *Son et al., 2012*), including condensed neuronal cell bodies and an accumulation of lipofuscin and vacuoles within spinal motor neurons (*Figure 2G* and *Figure 2—figure supplement 1E*). Similar pathologies were evident in neurons in the dorsal root ganglia, brain stem, and cerebellum (data not shown). These changes were accompanied by an overall reduction in neuronal density in the spinal cord (*Figure 2—figure supplement 1F*). Finally, *Norad*$^{-/-}$ mice showed decreased overall survival over a 2-year interval (*Figure 2H*). These findings demonstrate that *Norad* is essential to suppress widespread aging-associated degenerative phenotypes in mice.

## PUMILIO hyperactivity in *Norad*-deficient mice

Previous work established that *NORAD* is the preferred binding partner of PUM2 in human cells (*Lee et al., 2016*). *NORAD* knockout or knockdown triggers PUMILIO hyperactivity and a consequent loss of genomic stability due to excessive PUMILIO-mediated repression of a set of target mRNAs that are critical for normal mitosis (*Lee et al., 2016*; *Tichon et al., 2016*). Given the prior demonstration that genomic instability in mice causes aging-associated phenotypes (*Baker et al., 2004*; *Baker et al., 2006*), the regulation of PUMILIO activity by this lncRNA, and the resulting effects of PUMILIO hyperactivity, could potentially underlie the phenotype of *Norad*-deficient animals.

To assess whether *Norad* regulates PUMILIO activity in mice, we performed enhanced UV cross-linking immunoprecipitation coupled with high-throughput sequencing (eCLIP) (*Van Nostrand et al., 2016*) to assess the transcriptome-wide interactions of PUM2 with target RNAs in *Norad*$^{+/+}$ and *Norad*$^{-/-}$ mice (*Supplementary file 1*). Brain was chosen for these experiments because *Norad* shows the highest expression in this tissue (*Figure 1B* and *Figure 1—figure supplement 1B*), pathologic changes are present in the CNS of *Norad*$^{-/-}$ mice (*Figure 2G* and *Figure 2—figure supplement 1E–F*), and mammalian PUMILIO proteins have been implicated in various neuronal functions (*Driscoll et al., 2013*; *Gennarino et al., 2018*; *Gennarino et al., 2015*; *Siemen et al., 2011*; *Vessey et al., 2010*; *Vessey et al., 2006*; *Zhang et al., 2017*).

Like human *NORAD*, the mouse transcript is highly enriched for PREs, harboring 11 perfect matches to the canonical PRE consensus and an additional 3 PREs conforming to a slightly relaxed consensus sequence (UGUANAUN) (*Figure 3A*). Accordingly, robust binding of PUM2 to *Norad* was detectable by eCLIP, with the majority of binding occurring in the vicinity of PRE sequences.

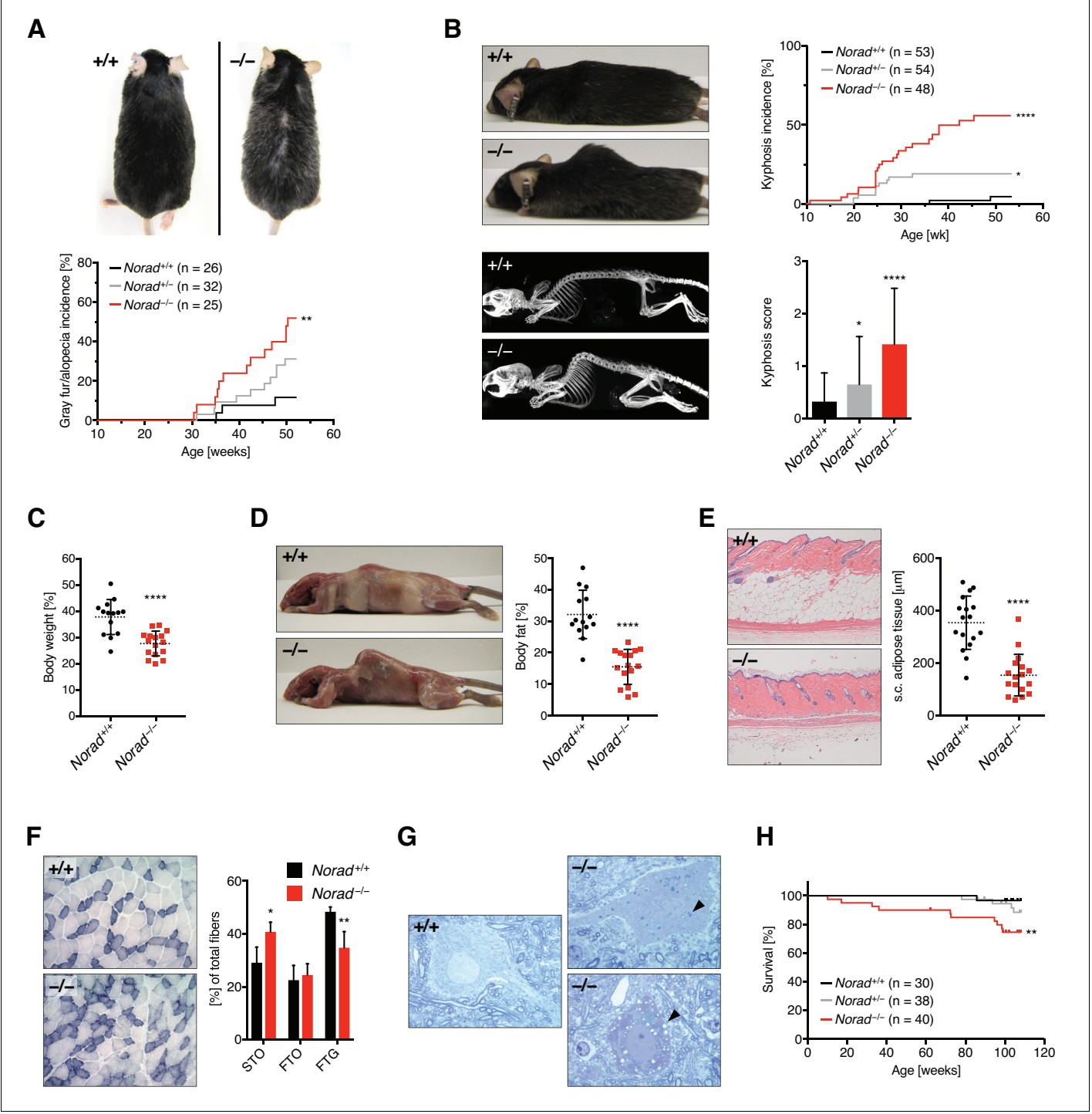

**Figure 2.** *Norad* loss-of-function results in a degenerative phenotype resembling premature aging. (**A**) Increased alopecia and gray fur in *Norad*−/− males. Representative 12-month-old male mice shown. (**B**) Kyphosis in *Norad*−/− mice. Kyphosis severity was scored from 0-3 using an established scheme (Guyenet et al., 2010). Upper right graph shows kyphosis incidence (score ≥ 2), lower right graph shows average kyphosis score at 12 months of age. Representative photographs and x-ray images of 12-month-old mice are shown. (**C**) Body weight of 12-month-old *Norad*−/− mice with a kyphosis score ≥ 2 compared to randomly-selected *Norad*+/+ controls (n = 14-16 mice per genotype). (**D**) Body fat percentage of mice from panel (**C**), quantified using NMR. Photographs show representative 12-month-old mice with skin removed to demonstrate loss of fat depots. (**E**) Subcutaneous (s.c.) fat thickness in 12-month-old *Norad*−/− mice with a kyphosis score ≥ 2 compared to randomly selected *Norad*+/+ controls (n = 17 mice per genotype). Representative H and E-stained skin sections shown. (**F**) Increased oxidative muscle fibers in 12-month-old *Norad*−/− mice. Fiber types in gastrocnemius muscle were grouped into slow twitch oxidative (STO), fast twitch oxidative (FTO), or fast twitch glycolytic (FTG) based on their high, intermediate or

*Figure 2 continued on next page*

*Figure 2 continued*

low SDH activity (n = 4 mice per genotype). Representative images of SDH histochemistry of gastrocnemius middle zones are shown. (**G**) Aging-related changes in the CNS of 12-month-old *Norad*$^{-/-}$ mice. Semi-thin sections of spinal cord demonstrate the presence of motor neurons with an accumulation of lipofuscin (arrowhead, upper right panel) or vacuoles (arrowhead, lower right panel) in *Norad*$^{-/-}$ mice. (**H**) Overall survival of mice of the indicated genotypes over a 2-year period. Data are represented as mean ± SD in (**B**)-(**F**), and p values were calculated using log-rank test in (**A**), (**B**), and (**H**) or Student's t test in (**B**)-(**F**). *p ≤ 0.05, **p ≤ 0.01, ****p ≤ 0.0001.

DOI: https://doi.org/10.7554/eLife.42650.006

The following figure supplement is available for figure 2:

**Figure supplement 1.** *Norad* loss-of-function results in a degenerative phenotype resembling premature aging.

DOI: https://doi.org/10.7554/eLife.42650.007

Strikingly, *Norad* was by far the most highly bound PUM2 target in the transcriptome (*Figure 3B*), exhibiting at least 1000 times greater CLIP signal than 95% of all PUM2-bound mRNAs. Thus, as observed in human cell lines, but to a much greater extent in vivo, *Norad* is the preferred RNA target of PUM2 in mouse brain.

We next examined the transcriptome-wide interactions of PUM2 with target mRNAs. Notably, the relaxed PRE consensus was the most enriched sequence motif detected in PUM2-bound mRNA 3′ UTRs, supporting the reliability of this eCLIP dataset (*Figure 3B*). To estimate the apparent PUM2 CLIP signal for each target in *Norad*$^{+/+}$ and *Norad*$^{-/-}$ brain, the average reads in CLIP clusters in a given target 3′ UTR normalized to the expression level of the target was determined for each condition, an approach used previously to estimate relative binding in CLIP data (*Bosson et al., 2014*). This analysis suggested that PUM2 target occupancy was significantly increased in *Norad*$^{-/-}$ brain, consistent with PUMILIO hyperactivity (*Figure 3C*). Further supporting this conclusion, RNA-seq revealed that expression of PUM2 CLIP targets was significantly decreased in *Norad*$^{-/-}$ brain (*Figure 3D*). Augmented repression of PUM2 CLIP targets was even more apparent when specifically examining targets whose PUM2 binding was measurably increased in the *Norad*$^{-/-}$ condition (*Figure 3E*). Together, these data strongly support a conserved function for *Norad* as a negative regulator of PUMILIO activity in vivo.

### *Norad* deficiency leads to genomic instability

To assess whether *Norad* loss of function results in genomic instability, DNA FISH was used to quantify the number of marker chromosomes in primary hematopoietic cells, a representative mitotic tissue. This analysis revealed a significant increase in aneuploid lymphocytes and splenocytes in 3-month and 12-month-old *Norad*$^{-/-}$ mice (*Figure 4A*). To directly determine whether loss of *Norad* results in mitotic abnormalities, MEFs were examined using DNA FISH and live cell imaging. In contrast to lymphocytes or splenocytes, we observed a high rate of polyploidization in all MEF lines tested, as previously reported (*Todaro and Green, 1963*) (*Figure 4—figure supplement 1A–B*). We therefore excluded tetraploid and octaploid cells from those scored as aneuploid (higher ploidy was rarely observed). Despite applying these stringent criteria, we detected a significant increase in aneuploidy in *Norad*$^{-/-}$ MEFs (*Figure 4—figure supplement 1A*). Moreover, time lapse microscopy revealed an increased occurrence of anaphase bridges and lagging chromosomes as *Norad*$^{-/-}$ MEFs underwent mitosis (*Figure 4B*). Thus, *Norad*-deficient mouse tissues and cells exhibit genomic instability.

In human cell lines with *NORAD* deficiency, genome instability has been linked to repression of a program of genes that are required for mitosis, DNA repair, and DNA replication due to PUMILIO hyperactivity (*Lee et al., 2016*). Consistent with these findings, RNA-seq revealed significant repression of a similar set of genes in *Norad*$^{-/-}$ spleens (*Figure 4C–D*). Moreover, brain PUM2 CLIP targets with two or more PREs in their 3′ UTRs were downregulated in *Norad*$^{-/-}$ spleens, providing evidence of PUMILIO hyperactivity in this tissue (*Figure 4E*). Taken together, these data support a conserved, essential role for the *Norad*-PUMILIO axis in the maintenance of genomic stability in mammals.

### Loss of *Norad* results in mitochondrial dysfunction

Phenotypic analyses of *Norad*-deficient mice unexpectedly revealed that in addition to genomic instability, widespread mitochondrial dysfunction was evident in knockout tissues. Overt mitochondrial abnormalities were observed in skeletal muscle of 12-month-old *Norad*$^{-/-}$ mice, including large

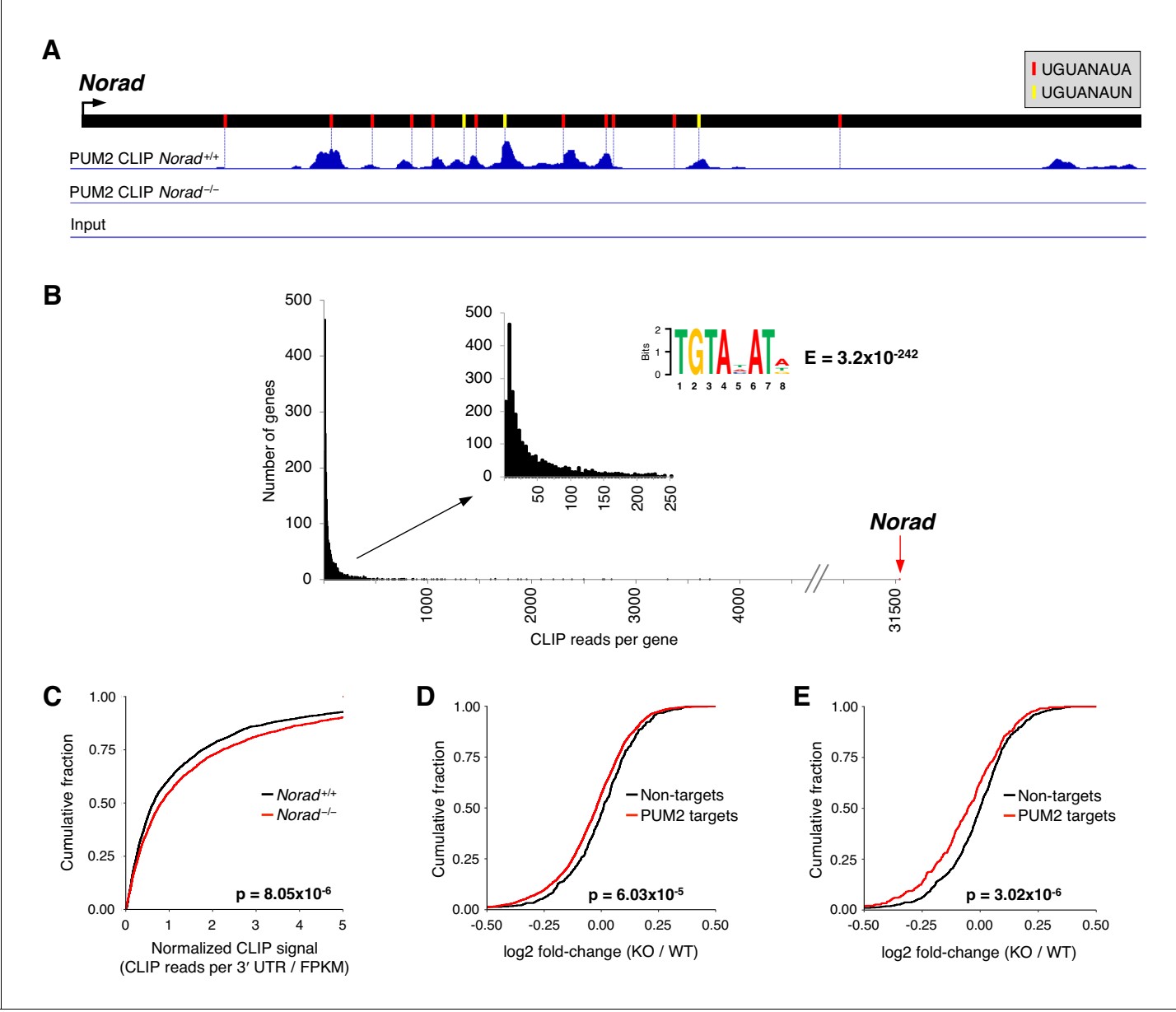

**Figure 3.** PUMILIO hyperactivity in *Norad*-deficient mice. (**A**) Normalized brain PUM2 CLIP reads mapped to *Norad* visualized using the Integrative Genomics Viewer (scale = 0–1260). Positions of perfect or relaxed PREs indicated with red or yellow lines, respectively. (**B**) *Norad* is the preferred PUM2 target RNA in mouse brain. The sum of all normalized reads in CLIP clusters in the 3' UTR of each PUM2 target RNA was calculated and data were plotted as a histogram showing numbers of genes with a given number of total CLIP reads. The web logo above the graphs shows the most significantly enriched motif identified by MEME-ChIP analysis (*Bailey et al., 2009*) in CLIP clusters in 3' UTRs. (**C**) Increased PUM2 target occupancy in *Norad*[−/−] brains. Cumulative distribution function (CDF) plot showing the normalized CLIP signal (total CLIP reads in clusters in the 3' UTR normalized to the RNA-seq-determined expression level of the gene) for PUM2 target genes detected in both *Norad*[+/+] and *Norad*[−/−] brains. (**D**) CDF plot comparing fold-changes of PUM2 CLIP targets detected in *Norad*[−/−] brains to non-targets with similar expression and 3' UTR lengths. Fold-changes calculated using *Norad*[−/−] vs. *Norad*[+/+] brain RNA-seq data. (**E**) As in (**D**) but only PUM2 CLIP targets with at least a 2-fold increase in normalized CLIP signal in *Norad*[−/−] vs. *Norad*[+/+] brain were plotted. p values calculated by Kolmogorov-Smirnov test for (**C**)-(**E**).

DOI: https://doi.org/10.7554/eLife.42650.008

accumulations of subsarcolemmal mitochondria (*Figure 5A–B*) accompanied by a significant increase in mitochondrial DNA (mtDNA) content (*Figure 5—figure supplement 1A*). Ultrastructurally, these mitochondria appeared irregular in shape and enlarged, with loss of cristae (*Figure 5B*). Similarly irregular and enlarged mitochondria were observed in spinal neurons of *Norad*[−/−] mice (*Figure 5C*).

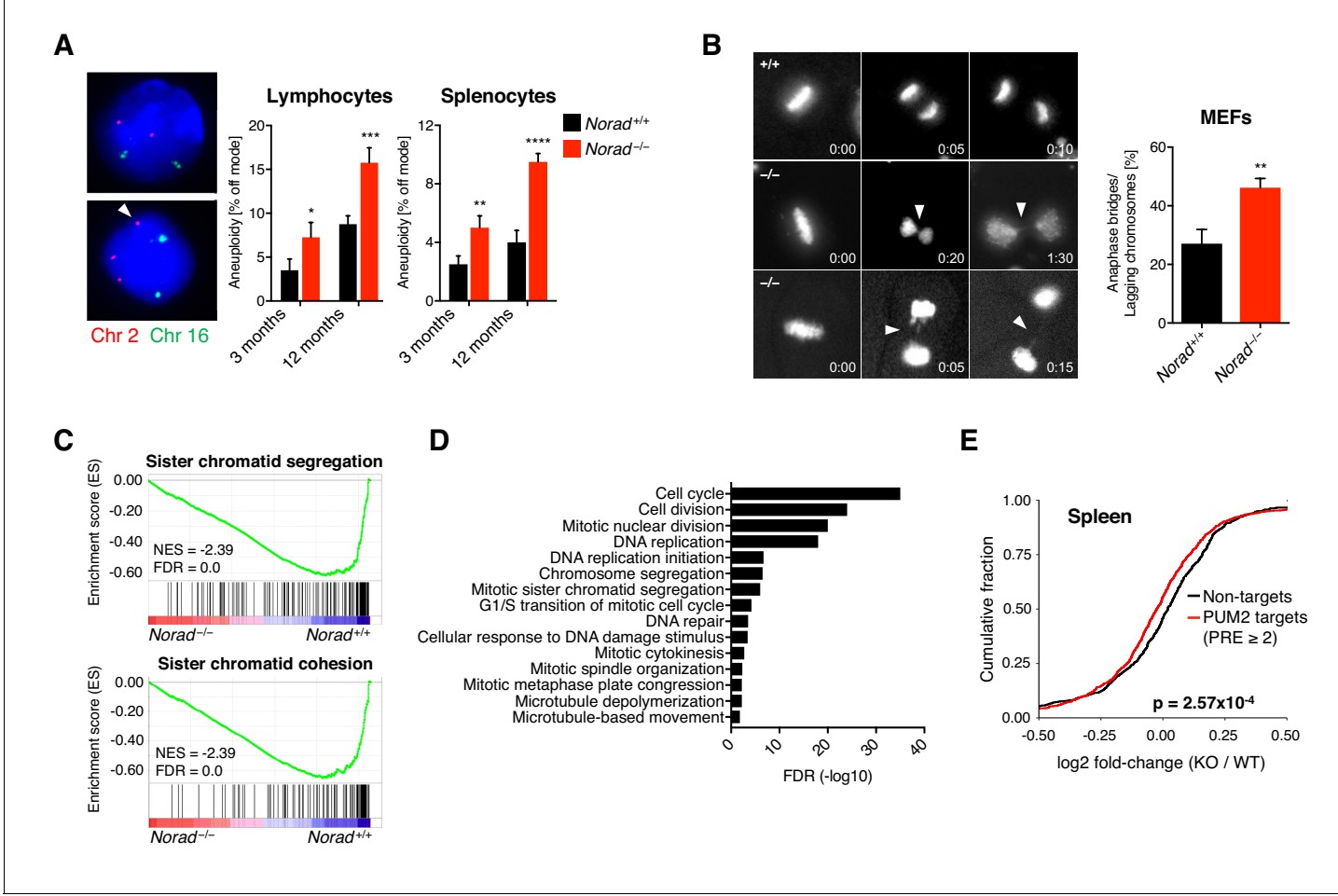

**Figure 4.** *Norad* deficiency leads to genomic instability. (**A**) DNA FISH for two representative chromosomes (chr. 2 and 16) was performed on cultured lymphocytes or freshly isolated splenocytes from mice of the indicated genotypes and ages. Frequency of cells with a non-modal number of chromosomes was determined by scoring 100 interphase nuclei per sample (n = 4 mice per genotype per time-point). Images show examples of on mode (upper panel) or off mode (lower panel) cells. (**B**) Representative time-lapse images of metaphase-to-anaphase transitions of Hoechst-stained primary MEFs show anaphase bridges and lagging chromosomes (arrowheads). Time stamp indicates hr:min elapsed. Graph represents data from a total of 128 *Norad*$^{+/+}$ and 158 *Norad*$^{-/-}$ mitoses analyzed in independent MEF lines per genotype. (**C**) GSEA showing repression of indicated gene ontology (GO) gene sets in RNA-seq data from *Norad*$^{-/-}$ spleens. FDR, false discovery rate; NES, normalized enrichment score calculated by GSEA algorithm (Subramanian et al., 2005). (**D**) GO analysis was performed using genes that were significantly downregulated in *Norad*$^{-/-}$ spleens (EdgeR p ≤ 0.05) using DAVID. Significantly enriched biological processes (BP) are depicted in the graph. (**E**) CDF plot comparing fold-changes of PUM2 CLIP targets detected in *Norad*$^{-/-}$ brains with at least two PREs and expressed at FPKM ≥ 1 in spleen to non-targets with similar expression and 3′ UTR lengths. Fold-changes calculated using *Norad*$^{-/-}$ vs. *Norad*$^{+/+}$ spleen RNA-seq data. Data are represented as mean ± SD in (**A**) and (**B**), and p values were calculated using Student's t test. *p ≤ 0.05, **p ≤ 0.01, ***p ≤ 0.001, ****p ≤ 0.0001. p value in (**E**) was calculated by Kolmogorov-Smirnov test.
DOI: https://doi.org/10.7554/eLife.42650.009

The following figure supplement is available for figure 4:

**Figure supplement 1.** *Norad* deficiency leads to genomic instability.
DOI: https://doi.org/10.7554/eLife.42650.010

These structural abnormalities were accompanied by evidence of reduced mitochondrial function, such as decreased cytochrome c oxidase (COX; also known as Complex IV of the electron transport chain) activity in spinal neurons (*Figure 5D*). Additionally, rare COX-negative fibers were observed in *Norad*-deficient but not wild-type skeletal muscle (*Figure 5—figure supplement 1B*). These findings were noteworthy given the extensive evidence linking a decrease in mitochondrial function to aging-associated phenotypes (*Sun et al., 2016*) and the previous demonstration that mice lacking proof-reading activity of the mtDNA polymerase, which consequently accumulate mtDNA mutations and

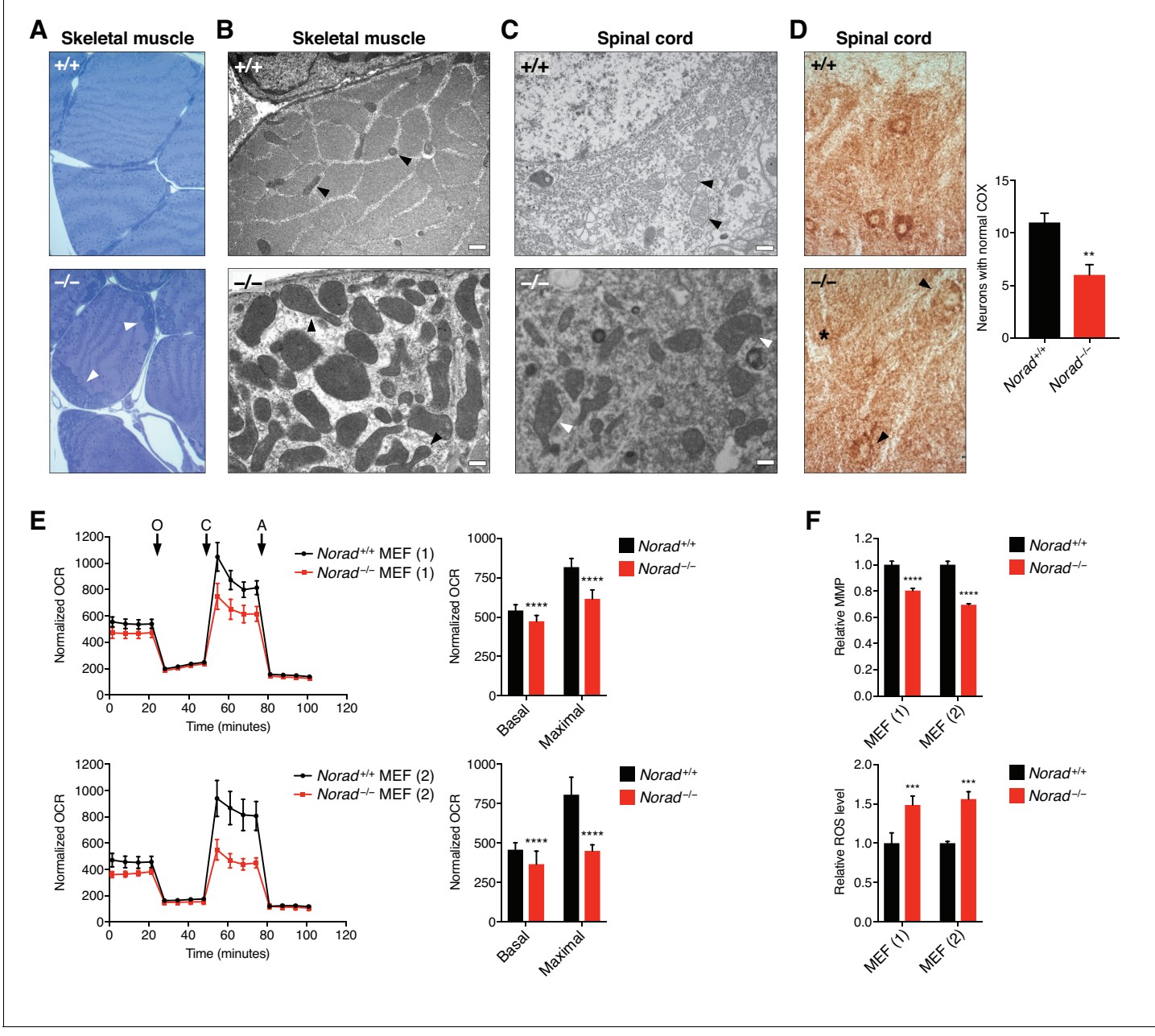

**Figure 5.** Loss of *Norad* results in mitochondrial dysfunction. (**A**) Subsarcolemmal accumulation of mitochondria (arrowheads) in skeletal muscle of 12-month-old *Norad*⁻/⁻ mice. Semi-thin sections of soleus muscle were stained with toluidine blue. (**B**)-(**C**) Electron micrographs of soleus muscle (**B**) and spinal motor neurons (**C**) showing mitochondrial morphology in 12-month-old *Norad*⁺/⁺ and *Norad*⁻/⁻ mice. Black or white arrowheads highlight mitochondria in each image. Scale bars 500 nm. (**D**) Reduced COX activity in the CNS of 12-month-old *Norad*⁻/⁻ mice. Spinal cord motor neurons were analyzed using COX histochemistry, and neurons with normal COX activity were counted (n = 7-8 sections total from 4 mice per genotype). Representative images are shown with arrowhead and asterisk pointing to neurons with decreased or absent COX activity, respectively. (**E**) Impaired respiration in immortalized *Norad*⁻/⁻ MEFs. Normalized oxygen consumption rates (OCR) (OCR/total protein) were determined in two littermate pairs of *Norad*⁺/⁺ and *Norad*⁻/⁻ MEFs using Seahorse analysis (n = 22-24 biological replicates per MEF pair, O = oligomycin, C = CCCP, A = antimycin A). Basal and maximal respiration was determined from measurement 4 and 12, respectively. (**F**) Reduced mitochondrial membrane potential (MMP) and elevated ROS levels in immortalized *Norad*⁻/⁻ MEFs. MMP and ROS levels were assessed in two littermate pairs of *Norad*⁺/⁺ and *Norad*⁻/⁻ MEFs using flow cytometry (n = 3 biological replicates per MEF pair). Data are represented as mean ± SD in (**D**)-(**F**), and p values were calculated using Student's t test. **p ≤ 0.01, ***p ≤ 0.001, ****p ≤ 0.0001.

DOI: https://doi.org/10.7554/eLife.42650.011

The following figure supplements are available for figure 5:

*Figure 5 continued on next page*

*Figure 5 continued*

**Figure supplement 1.** Loss of *Norad* results in mitochondrial dysfunction.
DOI: https://doi.org/10.7554/eLife.42650.012
**Figure supplement 2.** Loss of *NORAD* results in mitochondrial dysfunction and increased ROS levels in human HCT116 cells.
DOI: https://doi.org/10.7554/eLife.42650.013

deletions, exhibit a premature aging phenotype with many similarities to that seen in *Norad*$^{-/-}$ mice (*Trifunovic et al., 2004*).

A major consequence of mitochondrial dysfunction that is believed to play a role in cellular damage and aging is the accumulation of reactive oxygen species (ROS) (*Raha and Robinson, 2000*; *Zorov et al., 2014*). Indeed, brain and spinal cord of *Norad*$^{-/-}$ mice show evidence of oxidative damage, including elevated levels of 3-nitrotyrosine (3-NT), 4-hydroxynonenal (4-HNE), and 8-hydroxy-2'-deoxyguanosine/8-hydroxyguanosine (8-OHdG/8-OHG), markers of protein, lipid, and nucleic acid oxidation, respectively (*Figure 5—figure supplement 1C–D*).

To directly assess mitochondrial function in *Norad*-deficient cells, respiration rates were analyzed in pairs of littermate-matched *Norad*$^{+/+}$ and *Norad*$^{-/-}$ MEFs. Basal and maximal respiration was significantly reduced in *Norad*$^{-/-}$ cells (*Figure 5E*), accompanied by a decrease in mitochondrial membrane potential (MMP) and an increase in ROS production (*Figure 5F*). Respiration was also examined in human *NORAD*$^{-/-}$ HCT116 cells (*Lee et al., 2016*). Unlike MEFs, HCT116 cells lacking *NORAD* exhibited a significant increase in mitochondrial content (*Figure 5—figure supplement 2A–C*). Nevertheless, when normalized to mtDNA copy number, a similar reduction in respiration and increase in ROS was detectable in these cells (*Figure 5—figure supplement 2D–E*). These results document a previously unrecognized requirement for *Norad* in the maintenance of mitochondrial homeostasis in mammalian cells and tissues.

To investigate the mechanism through which *Norad* loss-of-function leads to mitochondrial dysfunction, we examined RNA-seq data from *Norad*$^{+/+}$ and *Norad*$^{-/-}$ brain and spleen using Gene Set Enrichment Analysis (GSEA) (*Subramanian et al., 2005*). Genes associated with mitochondria-related gene ontology (GO) terms, such as mitochondrial protein complex, electron transport chain, and oxidative phosphorylation, were significantly repressed in *Norad*-deficient tissues (*Figure 6—figure supplement 1A–B*). Remarkably, identical gene sets were repressed in human *NORAD*$^{-/-}$ HCT116 cells. We further identified a set of PUM2 brain CLIP targets within these gene sets that are known to perform important functions in mitochondrial biogenesis and homeostasis, mitochondrial transport, oxidative phosphorylation, metabolism, and ROS detoxification (*Figure 6A*). Downregulation of a representative set of these genes was validated by qRT-PCR in *Norad*$^{-/-}$ brain, spleen, and multiple independent MEF lines (*Figure 6B* and *Figure 6—figure supplement 1C*). These data are consistent with a model in which PUMILIO hyperactivity in *Norad*-deficient cells and tissues leads to coordinated downregulation of a broad set of target genes that are critical for normal mitochondrial function.

## Enforced PUM2 expression phenocopies *Norad* loss of function

While widespread genomic instability and mitochondrial dysfunction would be predicted to result in the premature aging-like phenotype displayed by *Norad*$^{-/-}$ mice, it remained to be demonstrated whether PUMILIO hyperactivity alone could account for the full spectrum of observed phenotypes. To address this question, transgenic mice with doxycycline (dox)-inducible expression of FLAG-tagged PUM2 were generated and crossed to mice harboring a ubiquitously expressed reverse tetracycline-controlled transactivator 3 transgene (*CAG-rtTA3*) (*Premsrirut et al., 2011*) (*Figure 7A*). Administration of dox induced broad transgene expression in *Pum2; rtTA3* double transgenic mice, as documented by FLAG immunohistochemistry (IHC) (*Figure 7—figure supplement 1A*). Robust transgenic PUM2 expression was also detectable by western blot in isolated MEFs (*Figure 7—figure supplement 1B*), although an increase in total PUM2 protein levels was surprisingly not detectable in bulk tissue (*Figure 7—figure supplement 1C*). Importantly, RNA-seq of spleens from transgenic animals revealed significant repression of PUM2 targets, demonstrating PUM2 hyperactivity in this tissue (*Figure 7—figure supplement 1D*). Because *CAG-rtTA3* does not efficiently drive transgene expression in the CNS (*Premsrirut et al., 2011*), we focused our phenotypic studies of *Pum2; rtTA3* double transgenic mice on peripheral tissues.

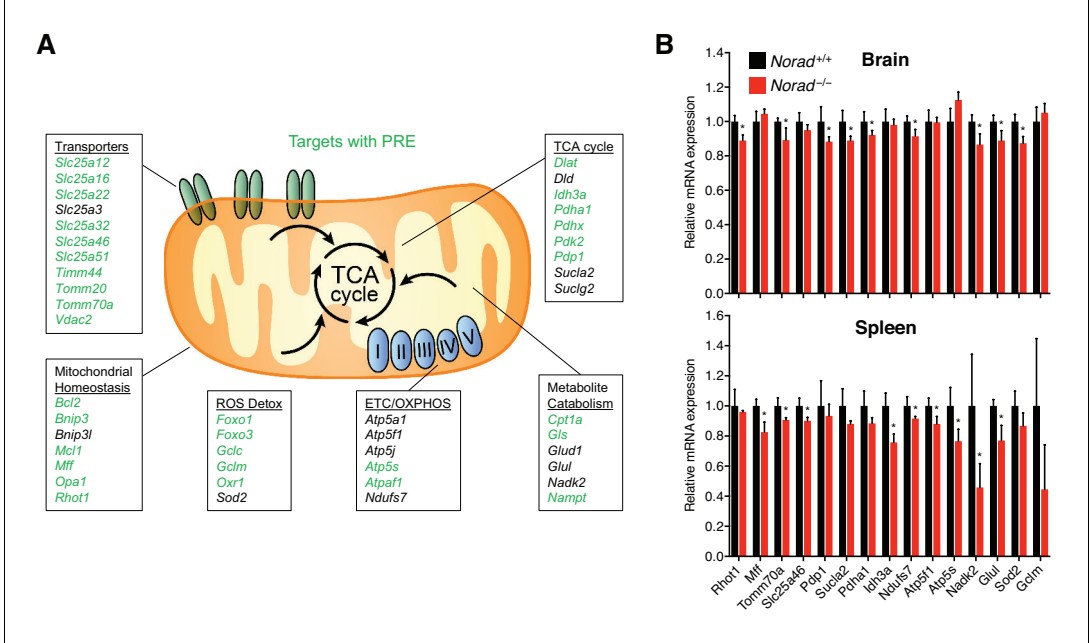

**Figure 6.** Loss of *Norad* results in the repression of key mitochondrial PUM2 target genes. (**A**) Selected PUM2 CLIP targets with important mitochondrial functions. Green text indicates the presence of a relaxed PRE (UGUANAUN) within 100 nucleotides of a 3' UTR PUM2 CLIP cluster. (**B**) Downregulation of mitochondrial PUM2 target genes in *Norad*$^{-/-}$ brain and spleen. Expression levels were determined by qRT-PCR (n = 4 mice per genotype). Data are represented as mean ± SD in (**B**), and p values were calculated using Student's t test. *p ≤ 0.05.

DOI: https://doi.org/10.7554/eLife.42650.014

The following figure supplement is available for figure 6:

**Figure supplement 1.** Repression of mitochondrial PUM2 target genes in *Norad*$^{-/-}$ cells and tissues.

DOI: https://doi.org/10.7554/eLife.42650.015

Administration of dox to young (8–14 week-old) *Pum2; rtTA3* double transgenic mice derived from two independent founders, but not to *Pum2* or *rtTA3* single transgenic controls, resulted in a striking phenotype within 2 months that closely resembled the appearance of *Norad*$^{-/-}$ mice at 1 year of age. Dox-treated *Pum2; rtTA3* mice developed rapidly progressing kyphosis, alopecia, graying of fur, and loss of body fat (***Figure 7B–C*** and ***Figure 7—figure supplement 2A***). These phenotypes were accompanied by increased aneuploidy in splenocytes (***Figure 7D***) and the accumulation of subsarcolemmal, irregularly shaped mitochondria lacking normal cristae in skeletal muscle (***Figure 7E–F***). Further demonstrating mitochondrial abnormalities, a global reduction in COX activity was observed in *Pum2; rtTA3* skeletal muscle (***Figure 7G*** and ***Figure 7—figure supplement 2B***) together with scattered necrotic and regenerating fibers (***Figure 7H*** and ***Figure 7—figure supplement 2C***).

Lastly, we directly assessed whether enforced PUM2 expression impairs mitochondrial function in MEFs and human cell lines. Transient expression of FLAG-PUM2 in MEFs or stable expression of either PUM1 or PUM2 in HCT116 significantly impaired respiration (***Figure 7I***, ***Figure 7—figure supplement 2D***, ***Figure 7—figure supplement 3A–C***). Overall, these data provide compelling evidence that PUMILIO hyperactivity in *Norad*-deficient animals results in genomic instability, mitochondrial dysfunction, and ultimately a multi-system degenerative phenotype resembling premature aging.

## Discussion

Although important roles for a growing number of lncRNA-encoding loci have been uncovered in development and disease states (***Anderson et al., 2016***; ***Arun et al., 2016***; ***Sauvageau et al., 2013***), definitive examples of noncoding RNA-mediated functions that are essential for mammalian physiology and maintenance of homeostasis across tissues are limited. Our studies of the murine *Norad* ortholog reported here unequivocally establish the importance of this lncRNA, and the tight

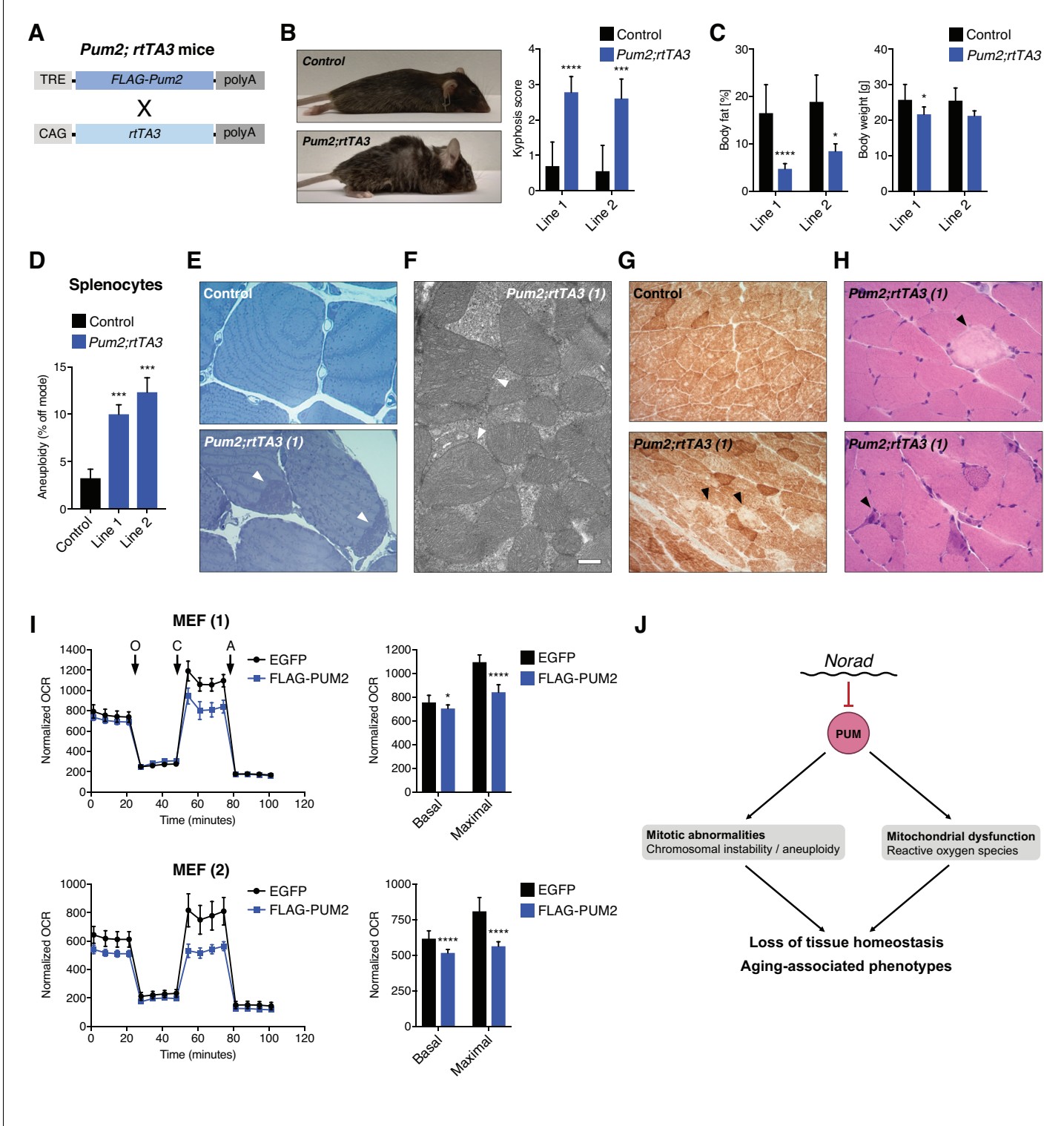

**Figure 7.** Enforced PUM2 expression phenocopies *Norad* loss of function. (**A**) Schematic depicting transgenes in doxycycline (dox)-inducible *Pum2* mice (TRE, TET responsive element; CAG, CAG promoter; rtTA3, reverse tetracycline-controlled transactivator 3; polyA, polyadenylation site). (**B**) Kyphosis in *Pum2; rtTA3* mice. Kyphosis scores were determined in two independent transgenic lines after 1.5-2 months of dox treatment. Dox-treated littermate matched wild-type and single transgenic *rtTA3* and *Pum2* mice were used as controls (n = 5-10 mice per genotype per transgenic line). (**C**) Reduced body fat and weight in *Pum2; rtTA3* mice after 1.5-2 months of dox treatment. Whole-body fat was quantified using NMR. Controls as in (**B**). (n = 7-12 (line 1) or 3-5 (line 2) mice per genotype). (**D**) Increased aneuploidy in *Pum2; rtTA3* splenocytes. DNA FISH was performed and quantified using freshly-isolated splenocytes after 2 months of dox treatment as in Figure 4A. Controls represent dox-treated single transgenic *rtTA3* and *Pum2* mice

*Figure 7 continued*

from line 1. 100 interphase nuclei per mouse were scored (n = 3-4 mice per genotype). (E) Subsarcolemmal accumulation of mitochondria (arrowheads) in skeletal muscle of *Pum2; rtTA3* (line 1) mice after 1.5 months of dox treatment. Representative dox-treated *Pum2* single transgenic control shown. Semi-thin sections of soleus muscle were stained with toluidine blue. (F) Representative electron micrograph showing abnormally enlarged mitochondria with distorted cristae (arrowheads) in skeletal muscle (soleus) of dox-treated *Pum2; rtTA3* (line 1) mice. Scale bar 500 nm. (G) COX histochemistry showing reduced COX activity in gastrocnemius muscle from a dox-treated *Pum2; rtTA3* (line 1) mouse compared to littermate-matched dox-treated *Pum2* single transgenic control. Arrowheads indicate muscle fibers with extreme reduction in COX activity. (H) Histologic analysis of H and E-stained sections of gastrocnemius muscle showing necrotic fibers (upper panel, arrowhead) and basophilic regenerating fibers (lower panel, arrowhead) in dox-treated *Pum2; rtTA3* (line 1) mice. (I) Impaired respiration in immortalized MEFs transiently overexpressing FLAG-PUM2. Normalized oxygen consumption rates (OCR) (OCR/total protein) were analyzed in two different *Norad*$^{+/+}$ MEF lines with either FLAG-PUM2 or EGFP overexpression using Seahorse analysis (n = 10-12 biological replicates per MEF line, O = oligomycin, C = CCCP, A = antimycin A). Basal and maximal respiration was determined from measurement 4 and 12, respectively. (J) Proposed mechanism for the aging-associated phenotypes associated with disruption of the *Norad*-PUMILIO axis. Data are represented as mean ± SD in (B)-(D) and (I), and p values were calculated using Student's t test. *p ≤ 0.05, ***p ≤ 0.001, ****p ≤ 0.0001.

DOI: https://doi.org/10.7554/eLife.42650.016

The following figure supplements are available for figure 7:

**Figure supplement 1.** Enforced PUM2 expression phenocopies *Norad* loss of function.
DOI: https://doi.org/10.7554/eLife.42650.017
**Figure supplement 2.** Enforced PUM2 expression phenocopies *Norad* loss of function.
DOI: https://doi.org/10.7554/eLife.42650.018
**Figure supplement 3.** PUMILIO overexpression results in mitochondrial dysfunction in human HCT116 cells.
DOI: https://doi.org/10.7554/eLife.42650.019

regulation of its target PUMILIO proteins, in mammalian biology and implicate the *Norad*-PUMILIO axis as a major regulator of aging-associated phenotypes (***Figure 7J***). These findings provide important new insights and open new lines of investigation into the roles of noncoding RNAs and RNA binding proteins in normal physiology, aging, and disease.

While PUMILIO proteins belong to a deeply conserved family of post-transcriptional regulators, obvious *Norad* orthologs are apparent only in mammals. Other RNAs that regulate the activity of PUMILIO-related proteins in a similar manner in non-mammalian species have not been reported. Why then did this additional layer of PUMILIO regulation evolve? A possible answer to this question may relate to recent findings that, together with those reported here, demonstrate an exquisite sensitivity to PUMILIO dosage in mammals. Zoghbi and colleagues recently showed that slightly reduced *PUM1* dosage causes neurodegeneration in human and mouse brain (***Gennarino et al., 2018***; ***Gennarino et al., 2015***). Human subjects carrying heterozygous *PUM1* deletions or missense mutations develop a neurodevelopmental disorder referred to as *PUM1*-associated developmental disability, ataxia, and seizure (PADDAS), associated with a ~ 50% reduction in PUM1 protein, or a later onset variant known as *PUM1*-related cerebellar ataxia (PRCA), associated with only a ~ 25% lowering of PUM1 levels (***Gennarino et al., 2018***). Taken together with our findings from this study, in which we examined for the first time the effects of mammalian PUMILIO hyperactivity in vivo, we can conclude that PUMILIO activity must be maintained within a very narrow range in order to prevent widespread deleterious consequences. In light of these findings, we propose that one major function of *Norad* is to buffer PUMILIO activity such that it stays within this critical range. This model posits the existence of *Norad*-bound and free PUMILIO pools which are exchangeable and in equilibrium, ensuring a consistent amount of available PUMILIO for target mRNA engagement and preventing fluctuations in PUMILIO expression from manifesting in altered target repression. Indeed, an RNA such as *Norad* represents an ideal molecule to serve as a buffer of this type, as it is able to efficiently regulate the activity of a pre-existing pool of PUMILIO at the level of target engagement.

In addition to providing a buffering function, it is likely that *Norad* is also utilized for dynamic regulation of PUMILIO activity under selected conditions. In particular, this lncRNA is known to be induced by a variety of cellular stressors, including DNA damage (***Lee et al., 2016***) and hypoxia (***Michalik et al., 2014***), which would be expected to result in de-repression of PUMILIO targets following these stimuli. Although the functional consequences of modulating PUMILIO-mediated gene regulation under these conditions is not yet understood, continued investigation of the signaling inputs that control this system and the resulting effects on PUMILIO-regulated gene networks will be important to further elucidate the roles of this newly discovered pathway in mammalian biology.

A surprising observation reported in this study was the rapid onset of dramatic premature aging-like phenotypes in *Pum2* transgenic mice despite a lack of overt overexpression of PUM2 protein at the bulk tissue level (*Figure 7—figure supplement 1C*). This finding raises the question of how transgene induction is able to drive such a striking phenotype if the protein product does not accumulate to supraphysiologic levels. An appealing hypothesis to explain this observation postulates that there are key vulnerable cell populations that become dysfunctional or damaged upon *Pum2* induction. These cells may be rare or may naturally express lower levels of PUMILIO, such that a large change in PUM2 expression within them may be masked by the majority of cells in the tissue. Indeed, isolated MEFs from *Pum2; rtTA3* transgenic mice show a robust increase in PUM2 protein expression (*Figure 7—figure supplement 1B*), thereby demonstrating clear transgene activity in isolated cell types. In addition, given that production of transgenic FLAG-PUM2 is robustly detectable by IHC (*Figure 7—figure supplement 1A*), repression of endogenous PUMILIO through a previously described negative feedback mechanism (*Galgano et al., 2008*; *Kedde et al., 2010*; *Morris et al., 2008*) may also contribute to the apparent lack of increase in total PUM2 levels in bulk tissue. Most importantly, the significant repression of PUM2 targets upon transgene induction (*Figure 7—figure supplement 1D*) demonstrates detectable PUM2 hyperactivity in *Pum2; rtTA3* tissues even in the absence of a global increase in protein abundance. Finally, it is worth noting that the absolute magnitude of repression of individual PUM2 targets in *Norad*$^{-/-}$ or *Pum2; rtTA3* tissues is generally low (~10–20%) (for example, see *Figure 6*). While the complexity of bulk tissue may mask more robust repression of PUM2 targets in specific cell-types, it is likely that the phenotypic consequences of PUMILIO hyperactivity cannot be attributed to repression of individual targets. Rather, the coordinated, modest repression of a broad set of PUMILIO targets in aggregate most likely produces the dramatic phenotypes observed.

Cells that are sensitive to enforced *Pum2* expression likely include stem cell populations whose dysfunction could lead to loss of tissue homeostasis and degenerative phenotypes. Accordingly, PUMILIO and related proteins have been implicated as critical stem cell regulators in model organisms and mammals (*Crittenden et al., 2002*; *Forbes and Lehmann, 1998*; *Leeb et al., 2014*; *Lin and Spradling, 1997*; *Naudin et al., 2017*; *Shigunov et al., 2012*). Identification of specific cell populations that drive aging-associated phenotypes under conditions of *Norad*-deficiency or PUMILIO hyperactivity represents an important priority for future work as this approach may reveal new cell types whose dysfunction contributes to the natural aging-associated decline of tissue homeostasis and renewal.

Analyses of *Norad*-deficiency and enforced *Pum2* expression unexpectedly revealed that PUMILIO hyperactivity triggers the coordinated repression of a large set of PUM2 target transcripts with key roles in mitochondrial function and homeostasis, associated with widespread structural and functional mitochondrial defects. These findings were corroborated by a recent study reporting that elevated PUM2 levels in aged mice impaired mitochondrial homeostasis (*D'Amico et al., 2019*). A large body of evidence has linked a decline in mitochondrial function to aging-associated phenotypes (*Sun et al., 2016*), including the direct demonstration that 'mitochondrial mutator mice', which harbor a mutation in the mitochondrial DNA polymerase and consequently accumulate mtDNA mutations, develop a premature aging phenotype with many features in common with *Norad*$^{-/-}$ mice (*Trifunovic et al., 2004*). Thus, mitochondrial dysfunction in concert with genomic instability, another abnormality associated with premature aging in mice (*Baker et al., 2004*; *Baker et al., 2006*), provides a compelling mechanistic basis for the phenotype of *Norad*-deficient animals. The regulation of mitochondrial biogenesis and function by PUMILIO-related proteins is not restricted to mammals. The budding yeast Puf family member Puf3p preferentially associates with mRNAs that encode mitochondrial proteins and facilitates their local translation in the vicinity of the mitochondrial protein import machinery (*García-Rodríguez et al., 2007*; *Gerber et al., 2004*; *Saint-Georges et al., 2008*). In *Drosophila* and cultured mammalian cells, PUMILIO proteins repress translation of mRNAs that encode mitochondria-destined proteins until these transcripts are docked at the mitochondrial outer membrane (*Gehrke et al., 2015*). Together, these observations suggest a deeply conserved role for PUMILIO proteins in the regulation of mitochondrial biology across eukaryotic species.

Perhaps, the most intriguing question to arise from these studies is whether dysregulation of the *Norad*-PUMILIO axis plays a role in physiologic aging and/or human disease. Remarkably, a recent RNA-seq study of noncoding RNA expression in the subependymal zone of human brains of increasing age reported a strong age-related decrease in *NORAD* expression (*Barry et al., 2015*). These

findings take on added significance in light of our new understanding of the consequences of disruption of the *Norad*-PUMILIO axis and suggest that this lncRNA, and its target PUMILIO proteins, represent new candidates whose altered expression or function may influence the normal age-related decline in tissue function. These genes also represent previously unrecognized candidates that may be mutated or otherwise disrupted in rare progeroid cases that are unlinked to the genes that are presently known to cause these disorders. Thus, further study of the *Norad*-PUMILIO axis, and the pathways that regulate this noncoding RNA and its target proteins, promises to reveal important and unexpected new insights into mammalian physiology and disease.

# Materials and methods

**Key resources table**

| Reagent type (species) or resource | Designation | Source or reference | Identifiers | Additional information |
|---|---|---|---|---|
| Gene (*Mus musculus*) | *Norad* (*2900097C17Rik*) | NA | Ensembl: ENSMUSG00000102869 | |
| Gene (*Homo sapiens*) | *NORAD* (*LINC00657*) | NA | Ensembl: ENSG00000260032 | |
| Genetic reagent (*Mus musculus*) | *Norad*<sup>-/-</sup> mice | This paper; see Materials and methods section Generation of mice with *Norad* deletion or enforced PUM2 expression | | |
| Genetic reagent (*Mus musculus*) | *Pum2* (TRE-*FLAG-Pum2*) mice | This paper; see Materials and methods section Generation of mice with *Norad* deletion or enforced PUM2 expression | | |
| Genetic reagent (*Mus musculus*) | *rtTA3* (CAG-*rtTA3*) mice | The Jackson Laboratory | Stock# 016532 | |
| Genetic reagent (*Mus musculus*) | C57BL/6J mice | UTSW Breeding Core | | |
| Cell line (*Mus musculus*) | *Norad*<sup>-/-</sup> primary MEFs | This paper; see Materials and methods section Generation and culture of mouse embryonic fibroblasts (MEF) | | |
| Cell line (*Mus musculus*) | *Norad*<sup>+/-</sup> primary MEFs | This paper; see Materials and methods section Generation and culture of mouse embryonic fibroblasts (MEF) | | |
| Cell line (*Mus musculus*) | *Norad*<sup>+/+</sup> primary MEFs | This paper; see Materials and methods section Generation and culture of mouse embryonic fibroblasts (MEF) | | |
| Cell line (*Mus musculus*) | *Norad*<sup>-/-</sup> immortalized MEFs | This paper; see Materials and methods section Generation and culture of mouse embryonic fibroblasts (MEF) | | |
| Cell line (*Mus musculus*) | *Norad*<sup>+/+</sup> immortalized MEFs | This paper; see Materials and methods section Generation and culture of mouse embryonic fibroblasts (MEF) | | |

*Continued on next page*

*Continued*

| Reagent type (species) or resource | Designation | Source or reference | Identifiers | Additional information |
|---|---|---|---|---|
| Cell line (*Mus musculus*) | *rtTA3* primary MEFs | This paper; see Materials and methods section Generation and culture of mouse embryonic fibroblasts (MEF) | | |
| Cell line (*Mus musculus*) | *Pum2* primary MEFs | This paper; see Materials and methods section Generation and culture of mouse embryonic fibroblasts (MEF) | | |
| Cell line (*Mus musculus*) | *Pum2;rtTA3* primary MEFs | This paper; see Materials and methods section Generation and culture of mouse embryonic fibroblasts (MEF) | | |
| Cell line (*Mus musculus*) | Neuro-2a | ATCC | CCL-131 | |
| Cell line (*Mus musculus*) | CT26 | ATCC | CRL-2638 | |
| Cell line (*Mus musculus*) | Hepa1-6 | ATCC | CRL-1830 | |
| Cell line (*Homo sapiens*) | HCT116 | ATCC | CCL-247 | |
| Cell line (*Homo sapiens*) | *NORAD*$^{-/-}$ HCT116 | *Lee et al. (2016)* | | |
| Cell line (*Homo sapiens*) | PUM1 OE HCT116 | *Lee et al. (2016)* | | |
| Cell line (*Homo sapiens*) | PUM2 OE HCT116 | *Lee et al. (2016)* | | |
| Antibody | Anti-PUM2 (polyclonal goat) | Santa Cruz | sc-31535 | eCLIP |
| Antibody | Anti-PUM2 (monoclonal rabbit) | Abcam | ab92390 | WB (1:4000) |
| Antibody | Anti-PUM1 (monoclonal rabbit) | Abcam | ab92545 | WB (1:4000) |
| Antibody | Anti-GAPDH (monoclonal rabbit) | Cell Signaling | #2118 | WB (1:5000) |
| Antibody | Anti-FLAG (polyclonal rabbit) | Cell Signaling | #2368 | WB (1:1000) |
| Antibody | IRDye 800CW anti-rabbit (donkey) | Licor | 925–32213 | WB (1:10000) |
| Antibody | Anti-FLAG M2 (monoclonal mouse) | Sigma-Aldrich | F1804 | IHC |
| Antibody | Anti-4-HNE (polyclonal rabbit) | Abcam | ab46545 | IHC |
| Antibody | Anti-DNA/RNA Damage (monoclonal mouse) | Abcam | ab62623 | IF |
| Antibody | Anti-Digoxigenin (monoclonal mouse) | Roche | 11333062910 | RNA FISH |
| Antibody | Anti-Mouse IgG, Cy3 (polyclonal goat) | EMD Millipore | AP124C | RNA FISH |
| Recombinant DNA reagent | pTRE-Tight-*FLAG-Pum2* | This paper; see Materials and methods section Generation of mice with *Norad* deletion or enforced PUM2 expression | | |

*Continued*

| Reagent type (species) or resource | Designation | Source or reference | Identifiers | Additional information |
|---|---|---|---|---|
| Recombinant DNA reagent | pCAG-*FLAG-Pum2* | This paper; see Materials and methods section FLAG-PUM2 overexpression in MEFs | | |
| Recombinant DNA reagent | pcDNA3-*EGFP* | This paper; see Materials and methods section FLAG-PUM2 overexpression in MEFs | | |
| Recombinant DNA reagent | pX330-U6-Chimeric_BB-CBh-hSpCas9 | Addgene | #42230 | |
| Sequence-based reagent | Mouse IDetect Point Probe Chr. 2 (red) | Empire Genomics | IDMP1002-R | DNA FISH |
| Sequence-based reagent | Mouse IDetect Point Probe Chr. 16 (green) | Empire Genomics | IDMP1016-1-G | DNA FISH |
| Sequence-based reagent | *Norad* (*2900097C17Rik*) TaqMan assay | Applied Biosystems | Mm04242407_s1 | qPCR |
| Sequence-based reagent | *NORAD* custom TaqMan assay | *Lee et al. (2016)* | N/A | qPCR |
| Commercial assay or kit | Total ROS ID Detection Kit | Enzo Life Sciences | ENZ-51011 | |
| Commercial assay or kit | 3-nitrotyrosine (3-NT) ELISA | Mybiosource | MBS262795 | |
| Chemical compound, drug | Tetramethylrhodamine ethyl ester (TMRE) | Enzo Life Sciences | ENZ-52309 | |
| Software, algorithm | Prism 7 | GraphPad Software | | |
| Software, algorithm | NGS QC Toolkit (v2.3.3) | *Patel and Jain, 2012* | | |
| Software, algorithm | Tophat2 (v2.0.12) | *Kim et al., 2013* | | |
| Software, algorithm | HISAT2 (v2.1.0) | *Pertea et al., 2016* | | |
| Software, algorithm | FeatureCount (v1.4.6) and (v1.6.0) | *Liao et al. (2014)* | | |
| Software, algorithm | EdgeR (v3.8.6) and (v3.24.0) | *Robinson et al. (2010)* | | |
| Software, algorithm | Stringtie (v1.2.2) | *Pertea et al., 2015* | | |
| Software, algorithm | BEDtools | *Quinlan and Hall (2010)* | | |
| Software, algorithm | Gene set enrichment analysis | *Subramanian et al. (2005)* | | |
| Software, algorithm | Database for Annotation, Visualization and Integrated Discovery (DAVID) | *Huang et al., 2009a* | | |

## Generation of mice with *Norad* deletion or enforced PUM2 expression

All animal protocols were approved by the Institutional Animal Care and Use Committee (IACUC) of The University of Texas Southwestern Medical Center (UTSW) and The Ohio State University, Nationwide Children's Hospital. Mice were maintained in regular housing with a 12 hr light/dark cycle and normal chow and water *ad libitum*. *Norad*$^{-/-}$ mice were generated in the UTSW Transgenic Core by injecting Cas9 mRNA (Sigma-Aldrich) together with two in vitro transcribed sgRNAs flanking the *Norad* locus into fertilized C57BL/6J oocytes as described (*Yang et al., 2013*). Founder mice harboring deletions of *Norad* were maintained by backcrossing to wild-type C57BL/6J mice. Of note, *Norad*$^{-/-}$ lines were produced from three independent founder mice with *Norad* deletions (**Figure 1—figure supplement 2A**). All were phenotypically indistinguishable and used for subsequent studies of *Norad* function. For all animal experiments, cohort sizes were not pre-determined using

power calculations. Sufficiently large sample sizes were used to allow detection of statistically significant differences between experimental cohorts.

Doxycycline (dox)-inducible *FLAG-Pum2* transgenic mice (in this study referred to as *Pum2* mice) were generated in a C57BL/6J background by the UTSW Transgenic Core using standard procedures for pronuclear injection. For the generation of the transgene vector, a cDNA clone of isoform 3 of the mouse *Pum2* coding sequence (BC041773) was purchased from transOMIC Technologies, verified by Sanger sequencing, PCR amplified with primers adding a FLAG tag to the protein N-terminus, and cloned into the pTRE-Tight vector (Clontech). Of note, isoform 3 encodes for the shorter PUM2 variant, which can be detected as the lower of two PUM2 bands in western blots. Broad endogenous expression of this isoform was detected by PCR in all tested mouse tissues (data not shown), confirming its physiologic relevance. Transgene positive *Pum2* mice were crossed to a ubiquitous cytomegalovirus early enhancer element chicken beta-actin (CAG) promoter-driven reverse tetracycline-controlled transactivator 3 (*rtTA3*) mouse line, which was generated in the Lowe laboratory (*Premsrirut et al., 2011*) and obtained from The Jackson Laboratory (stock number 016532). The resulting *Pum2; rtTA3* double transgenic mice were used for subsequent experiments together with *Pum2*, *rtTA3*, and wild-type littermates as controls. Transgene expression was induced in 8–14 week-old mice for 1.5–2 months by administering 2 g/L doxycycline hydrochloride (dox) (Sigma-Aldrich) supplemented with 10 g/L sucrose (Research Products International) in drinking water.

## Isolation of lymphocytes and splenocytes

Mice were anesthetized with isoflurane (Henry Schein Animal Health) and subjected to retro-orbital bleeding. Approximately 200 μL of blood were collected, immediately heparinized with 500 USP units/mL (Fresenius Kabi), transferred to 1.3 mL of PB Max Karyotyping Medium supplemented with 50 μg/mL lipopolysaccharide (Gibco and Sigma-Aldrich), and incubated at 37°C for 48 hr with shaking. After this incubation, cells were harvested and processed for DNA FISH analysis. For the isolation of splenocytes, mice were euthanized with an overdose of isoflurane, and spleens were resected, minced with a razor blade in 1X Hank's balanced salt solution (HBSS) without calcium and magnesium (Gibco), passed through a 70 μm cell strainer (Corning), and washed with 1X phosphate buffered saline (PBS) (Sigma-Aldrich). Immediately after isolation, splenocytes were processed for DNA FISH analysis.

## Generation and culture of mouse embryonic fibroblasts (MEF)

*Norad*[+/–] females were bred to *Norad*[+/–] males or *Pum2* females were bred to *rtTA3* males and euthanized at embryonic day E14.5. Under sterile conditions, the uterine horns were removed and washed once with 70% ethanol and three times with 1X PBS. The embryos were then released and the heads and all visceral organs removed. The remainder of the embryo was finely minced using razor blades and treated with 0.25% trypsin/EDTA (Gibco) at 37°C for a total of 20 min. MEF growth medium consisting of DMEM with 4.5 g/L glucose, L-glutamine and sodium pyruvate (Gibco) supplemented with 1X nonessential amino acids (NEAA), 1X Antibiotic-Antimycotic (AA) (all Gibco), and 10% fetal bovine serum (FBS) (Gibco, Sigma-Aldrich) was added to inactivate the trypsin. Tissue chunks were disrupted by vigorous pipetting, centrifuged, resuspended in MEF growth medium, and plated in T25 flasks. The next day, non-adherent tissue debris was used for genotyping, while attached cells were transferred to a fresh tissue culture dish and designated as primary MEF passage 1 (P1). Aliquots of primary MEF P1 were frozen and stored until needed. Primary MEFs were used for a maximum of passages. Immortalized MEF lines were generated by transfecting primary MEFs with pSG5-SV40-Large-T-Antigen using Lipofectamine 3000 (Invitrogen) according to the manufacturer's instructions. Starting at 48 hr post transfection, cells were serially passaged 1:10 to select for SV40-immortalized MEFs. After 6 passages, all SV40 large T antigen-transfected MEF lines were regarded as immortalized and this passage was designated as immortalized MEF P1. Of note, the genders of the MEF lines used in this study are not known. All MEF lines were tested and confirmed to be mycoplasma free.

## Culture of established cell lines

All established mouse cell lines (Neuro-2a, CT26, and Hepa1-6) were obtained from ATCC and cultured in DMEM with 4.5 g/L glucose, L-glutamine and sodium pyruvate (Gibco) supplemented with

1X AA (Gibco), and 10% FBS (Sigma-Aldrich). The male colon cancer cell line HCT116 was obtained from ATCC (CCL-247) and cultured in McCoy's 5a medium (Gibco) supplemented with 1X AA (Gibco) and 10% FBS (Gibco, Sigma-Aldrich). The cell line was authenticated by ATCC using short tandem repeat (STR) analysis in November 2017. The generation of HCT116 *NORAD*$^{-/-}$ clones via transcription activator-like effector nuclease (TALEN)-mediated insertion of a Lox-Stop-Lox cassette as well as HCT116 PUM1 and PUM2 overexpression clones using lentiviral transduction was described previously (*Lee et al., 2016*). Cell lines were tested and confirmed to be mycoplasma free.

## RNA isolation and quantitative reverse transcription PCR (qRT-PCR)

Total RNA was isolated from cells or tissues using the miRNeasy Mini Kit (Qiagen) following the manufacturer's instructions including an on-column DNAse I digest to remove genomic DNA contamination. Complementary DNA (cDNA) was generated from 1 µg of total RNA using the SuperScript III First-Strand Synthesis SuperMix for qRT-PCR (Invitrogen) according to the manufacturer's protocol. Relative *Norad* expression in mouse was quantified using the Applied Biosystems TaqMan assay for *2900097C17Rik* (*Norad*) (Mm04242407_s1) and the TaqMan Universal II Master Mix (Applied Biosystems). Human *NORAD* was quantified using a custom TaqMan assay described previously (*Lee et al., 2016*). For all other genes analyzed in this study, expression was quantified using the Power SYBR Green PCR Master Mix (Applied Biosystems) together with primers provided in *Supplementary file 2*. RNA expression levels were normalized to 18S ribosomal RNA (cell line studies) or *Gapdh* mRNA (in vivo studies) using either standard curves of each gene or the comparative ΔCt method. The number of biological replicates is stated in the figure legends, each biological replicate was run with three technical replicates.

## Next-generation RNA sequencing (RNA-seq)

Total RNA was isolated from brains and spleens of 10-week-old male *Norad*$^{+/+}$ and *Norad*$^{-/-}$ mice (three mice per genotype) as well as from spleens of 16-week-old *Pum2; rtTA3* and control (*Pum2* and wild-type) mice after 4 weeks of dox treatment (four mice per group) using the miRNeasy Mini Kit (Qiagen) including a DNAse I digestion step to remove genomic DNA. RNA integrity was determined with the Agilent 2100 Bioanalyzer, and only RNA samples with an RNA integrity number (RIN) of greater than eight were used for subsequent analysis. Sequencing libraries were prepared with the TruSeq Stranded mRNA Library Prep Kit (Illumina) and sequenced using the 75 base pair (bp) single-read protocol on a NextSeq 500 platform (Illumina). Library prep and RNA-seq were performed by the UTSW McDermott Center Next-Generation Sequencing Core.

For RNA-seq in *Norad*$^{+/+}$ and *Norad*$^{-/-}$ mice, quality assessment of the sequencing data was performed with NGS QC Toolkit (v2.3.3) (*Patel and Jain, 2012*). Reads with more than 30% of nucleotides with a Phred quality score of less than 20 were removed from further analysis. Quality-filtered reads were then aligned to the mouse reference genome GRCm38 (mm10) using Tophat2 (v2.0.12) with default settings (*Kim et al., 2013*). Only reads uniquely mapped to the genome were kept for future analysis. Aligned reads were counted per gene ID using featureCount (v1.4.6) (*Liao et al., 2014*). Differential gene expression analysis was carried out using the R package EdgeR (v3.8.6) (*Robinson et al., 2010*). For RNA-seq in *Pum2; rtTA3* and control mice, reads were aligned to GRCm38 (mm10) using HISAT2 (v2.1.0) (*Pertea et al., 2016*). Only reads uniquely mapped to the genome were kept for future analysis. Aligned reads were counted per gene ID using featureCount (v1.6.0). Differential gene expression analysis was carried out using EdgeR (v3.24.0). For each comparison, genes were required to have at least one read in at least one sample to be considered as expressed. Differential gene expression analysis was performed using the GLM approach following EdgeR's official documentation. CPM (counts per million) and FPKM (fragments per kilobase million) were obtained using EdgeR and Stringtie (v1.2.2) (*Pertea et al., 2015*), respectively.

Gene set enrichment analysis (GSEA) (*Subramanian et al., 2005*) was performed using default gene sets of gene ontology (GO) terms. The results obtained from the RNA-seq analyses of brain and spleen (normalized reads as CPM) were used as input data. The normalized enrichment scores (NES) as well as the false discovery rates (FDR) are provided in the figures. GO analysis of spleen RNA-seq data was also carried out on genes that were significantly (p≤0.05) downregulated in

*Norad⁻/⁻* spleens using the Database for Annotation, Visualization and Integrated Discovery (DAVID) (*Huang et al., 2009a*; *Huang et al., 2009b*).

## Enhanced UV crosslinking immunoprecipitation (eCLIP)

PUM2 RNA interactions in the mouse brain were determined by eCLIP, following a previously published protocol (*Van Nostrand et al., 2016*). In brief, brains from 3-month-old *Norad⁺/⁺* and *Norad⁻/⁻* females (two mice per genotype representing two biological replicates) were resected and cut in halves. Per sample, one half of a brain was minced with razor blades in 1X ice-cold diethyl pyrocarbonate (DEPC)-treated PBS. The resulting tissue suspensions were UV crosslinked on ice in a Spectrolinker XL-1500 (Spectronics) at 254 nm three times at 400mJ/cm². The UV crosslinked tissues were centrifuged, snap-frozen in ethanol/dry ice, and stored at −80°C until needed. For the immunoprecipitation of PUM2, Protein G Dynabeads (Invitrogen) were used together with the same polyclonal goat anti-PUM2 antibody (sc-31535, Santa Cruz) used previously for PUM2 CLIP analysis in human cells (*Lee et al., 2016*). For each genotype, duplicate size-matched input and immunoprecipitation samples were prepared (four samples per genotype). In contrast to the original protocol (*Van Nostrand et al., 2016*), in which libraries were designed for paired-end sequencing, we adapted the RNA and DNA linker sequences for single-read sequencing. For each sample, separate sequencing libraries were generated using a unique modified RiL19-new RNA linker as well as a modified rand103Tr3-new DNA linker and AR17-new reverse transcription primer (sequences provided in *Supplementary file 2*). PCR library amplification was performed with polyacrylamide gel electrophoresis (PAGE)-purified oligonucleotides containing specific indexes (D501-D504, D701-D703). Single-read sequencing was performed on a NextSeq 500 platform using a NextSeq 500/550 High Output v2 Kit, 75 cycles (Illumina) in the UTSW McDermott Center Next-Generation Sequencing Core.

All adapter sequences were removed using Cutadapt with an e-value set to 0.1. All reads less than 18 nt after adapter trimming were discarded, and the unique molecular identifiers (10 nt randomers) for PCR duplication identification were trimmed and recorded using in-house scripts. Because of the high number of *NORAD* pseudogenes in the human and mouse genome, we followed a similar mapping strategy to that used in our previous study (*Lee et al., 2016*). Reads were first mapped to *Norad* (*2900097C17Rik*) before all remaining reads, which did not align to *Norad*, were mapped to GRCm38 (mm10) using Tophat2 (v2.0.12) with default settings (*Kim et al., 2013*). Only uniquely mapped genomic reads were retained. PCR duplicates were then removed based on the unique molecular identifier information using in-house scripts. All remaining reads were regarded as usable reads and subjected to cluster calling.

For each IP sample, the read coverage of each nucleotide was calculated and all regions with coverages of equal or greater than three were kept as candidate bins. The read counts of each IP/input pair were obtained for every bin, requiring at least a 50% sequence overlap. The fold-changes of the normalized read counts were then calculated for each bin: normalized fold-change = ((reads_in_bin[IP]+1)/total_usable_reads[IP])/((reads_in_bin[input]+1)/total_usable_reads[input]). Bins with fold-changes greater than or equal to four were considered as clusters. Finally, we filtered clusters for those that were detected in both biological replicates of either genotype. Clusters that overlapped with at least 30% of their length were merged. Bedgraph files of each sample were generated with BEDtools (*Quinlan and Hall, 2010*) using reads normalized to the total usable genomic read count and visualized with the Integrative Genomics Viewer (IGV). Genes with one or more CLIP clusters in their 3' UTRs were regarded as PUM2 target genes. The number of CLIP reads per gene (*Figure 3B*) was determined by calculating the weighted sum of all reads within 3' UTR CLIP clusters for each PUM2 target gene from both *Norad⁺/⁺* CLIP replicates. To calculate normalized CLIP signal of PUM2 targets in *Norad⁺/⁺* and *Norad⁻/⁻* brains (*Figure 3C*), a method similar to that used by Bosson *et al.* was used (*Bosson et al., 2014*). Genes were first filtered for those with an average expression level of at least 1 FPKM in brain RNA-seq data. The average normalized number of CLIP reads from both CLIP replicates within 3' UTR clusters of each PUM2 target gene were then summed and divided by the gene's expression level (average FPKM) in the respective genotype. For CDF plots depicting fold-changes of CLIP targets in brain (*Figure 3D–E*), non-CLIP targets were filtered for those whose expression levels were within 25% of the log mean value for this parameter in the set of CLIP targets. Non-CLIP targets were also filtered for those whose 3' UTR length was within 25% of the mean of 3' UTR lengths in the set of CLIP targets. For similar CDF plots in which the expression of CLIP targets

was examined in spleen (*Figure 4E* and *Figure 7—figure supplement 1D*), brain CLIP targets were filtered for those with FPKM $\geq$ 1 in either *Norad$^{-/-}$* or *Pum2; rtTA3* spleen and the presence of at least two PREs within 100 nucleotides of CLIP clusters in the target 3' UTR. Non-CLIP targets were filtered as described above for brain to match expression level and 3' UTR length to the set of CLIP targets shown in each plot.

## Quantification of mitochondrial DNA

Total DNA was isolated using the DNeasy Blood and Tissue Kit (Qiagen) according to the manufacturer's instructions. Mitochondrial (mtDNA) and nuclear (nDNA) DNA were quantified by qPCR using either human or mouse specific primers (*Supplementary file 2*) and the Power SYBR Green PCR Master Mix (Applied Biosystems). The quantity of mtDNA was then normalized to the quantity of nDNA. Both mtDNA and nDNA concentrations were determined using standard curves. The number of replicates is provided in the respective figure legends.

## Subcellular fractionation

Mouse cell lines (immortalized MEFs, Neuro-2a, CT26, or Hepa1-6) were seeded in triplicate and harvested the next day for subcellular fractionation, which was performed as previously described (*Lee et al., 2016*). Briefly, cell pellets were lysed in RLN1 buffer (50 mM Tris-HCl pH 8.0, 140 mM NaCl, 1.5 mM MgCl$_2$, 0.5% NP-40, RNAse inhibitor), incubated on ice for 5 min, and centrifuged. The supernatant contained the cytoplasmic fraction, while the pellet contained the nuclear fraction. Both fractions were then subjected to RNA isolation and equal cell equivalents of nuclear and cytoplasmic RNA were used in subsequent qRT-PCR reactions. All samples were tested for *Norad* as well as for *Neat1* (nuclear control) and *Actb* (cytoplasmic control). Because equal cell equivalents of nuclear and cytoplasmic RNA were used in each reaction, the sum of the expression level of each transcript in the nucleus plus cytoplasm can be set to 100%, thereby allowing determination of the percentage of each transcript localized to each compartment. *Neat1* and *Actb*, respectively, showed the expected nuclear and cytoplasmic localization in each experiment, confirming successful subcellular fractionation.

## Fluorescent in situ hybridization (RNA FISH)

RNA FISH was performed as described previously (*Mito et al., 2016*). A DIG-labeled RNA probe for mouse *Norad* was synthesized by in vitro transcription using a DIG-labeling mix (Roche). Primers used for amplification of the DNA template are provided in *Supplementary file 2*. MEFs grown on poly-L-lysine coated coverslips were fixed in 4% paraformaldehyde for 10 min followed by permeabilization in 0.5% Triton X-100 for 10 min. Samples were then hybridized with 10 ng/μL DIG-labeled RNA probe at 55°C for 16 hr. Following hybridization, samples were washed and treated with RNase A. DIG-labeled probes were detected using a mouse monoclonal anti-DIG primary antibody (Roche) and a Cy3-labeled goat anti-mouse IgG secondary antibody (EMD Millipore). A Zeiss LSM700 confocal microscope was used for imaging.

## Western blots

Cell and tissue lysates were prepared in RIPA buffer (50 mM Tris-HCl pH 8.0, 150 mM NaCl, 1% NP-40, 0.5% sodium deoxycholate, 0.1% sodium dodecyl sulfate) supplemented with cOmplete Protease Inhibitor Cocktail (Roche). Western blots were probed with monoclonal rabbit anti-PUM2 antibody (ab92390, Abcam), monoclonal rabbit anti-PUM1 antibody (ab92545, Abcam), polyclonal rabbit anti-FLAG (2368, Cell Signaling), or monoclonal rabbit anti-GAPDH antibody (2118, Cell Signaling). Bands were visualized using an IRDye 800CW donkey anti-rabbit IgG secondary antibody (925–32213, Licor) and an Odyssey CLx Imager (Licor).

## FLAG immunohistochemistry

Tissues were harvested from mice that had been treated with dox for 3.5–6.5 weeks. All tissues were fixed in 10% neutral buffered formalin (NBF) for 24–48 hr. Fixed samples were processed, paraffin embedded, and sectioned using standard procedures. To detect the expression of the *FLAG-Pum2* transgene, immunohistochemistry (IHC) was performed by the UTSW Tissue Management Shared

Resource using the monoclonal anti-FLAG M2 antibody (F1804, Sigma-Aldrich). Images were acquired on an AxioObserver Z1 microscope (Zeiss).

## Histologic analysis of skeletal muscle and the central nervous system (CNS)

12-month-old $Norad^{+/+}$ and $Norad^{-/-}$ mice were used for semi-thin and ultrastructural analysis. For these studies, one group of mice were given xylazine/ketamine anesthesia and euthanized by cardiac perfusion with 4% paraformaldehyde followed by 5% glutaraldehyde (both in 0.1 M phosphate buffer). Tissue samples from brain and spinal cord were removed under a dissecting microscope. A second group of mice were perfused with 4% paraformaldehyde and their muscles were removed and further fixed in situ in 5% glutaraldehyde. These tissues were dissected into small blocks and processed for plastic embedding using standard methods (Sahenk and Mendell, 1979). Thick (1 µm) sections were stained with toluidine blue and selected blocks were sectioned and examined with an electron microscope (Hitachi H7650). Brain and spinal cord segments were placed in 10% NBF and processed for paraffin embedding. Brain, spinal cord, and skeletal muscle were collected from additional mice for cryostat sectioning. Muscle tissues from $Pum2; rtTA3$ as well as $Pum2$ and $rtTA3$ single transgenic littermates were collected and processed as for $Norad^{-/-}$ mice. All transgenic mice were between 4 and 6 months old and had been treated with dox for 1.5–2 months.

To analyze neuronal cell loss in the ventral horn neuron pools, 5 µm thick, paraffin embedded and hematoxylin and eosin (H and E) stained lumbar spinal cord sections from $Norad^{+/+}$ (n = 5) and $Norad^{-/-}$ (n = 5) mice were analyzed. Motor neuron pools in the anterior horn cell areas of both hemicords from each section were included. For each mouse, sections from 1-2 levels were photographed at 20X magnification. Only cell bodies clearly showing a nucleolus on the plane of the section were considered. Equal numbers of spinal cord levels were analyzed in each group. Mean neuronal densities of anterior horn areas per lumbar cord level were calculated.

Succinate dehydrogenase (SDH) enzyme histochemistry was used to assess metabolic fiber type changes in the aging muscle. For this purpose, muscle fiber types were grouped into three categories: slow twitch oxidative (STO), fast twitch oxidative (FTO), and fast twitch glycolytic (FTG). Twelve-µm-thick cross-sections from the gastrocnemius muscles of 12-month-old $Norad^{+/+}$ and $Norad^{-/-}$ mice (n = 4 in each group, 2 males and 2 females) were stained for SDH activity. Three images, each representing a distinct zone of the gastrocnemius muscle (a deep zone predominantly composed of STO, an intermediate zone showing a checkerboard appearance of STO, FTO, or FTG, and a superficial zone predominantly composed of FTG fibers), were taken along the midline axis at 20X magnification using an Olympus BX41 microscope. Muscle fiber types were determined and counted based on their dark (STO), intermediate (FTO), or light (FTG) SDH staining as previously reported (Yalvac et al., 2017).

Mitochondrial function was assessed in spinal cord neurons of 12-month-old $Norad^{+/+}$ and $Norad^{-/-}$ mice using cytochrome c oxidase (COX) histochemistry. Twelve µm thick frozen lumbar spinal cord tissue sections were mounted onto superfrost glass slides (Thermo Fisher) and dried at room temperature for 1 hr before the COX enzymatic activity assay was performed. Two sections, 100 µm apart, were analyzed from each mouse. The anterior horn areas of each section were photographed at 10X magnification and the number of neurons with normal COX activity were determined. Counts from the hemicord anterior horn cell area with the higher number of neurons with normal COX activity was included in the analysis. Only cell bodies with an obviously visible nucleolus on the plane of the section and with a COX activity clearly above the background of gray matter were considered. The number of neurons with normal COX activity was counted per each section analyzed. COX activity was also qualitatively analyzed in fresh frozen gastrocnemius muscle sections from 12-month-old $Norad^{+/+}$ and $Norad^{-/-}$ mice and 4-6 month-old $Pum2; rtTA3$ transgenic mice or controls that had been treated with dox for 1.5–2 months.

## DNA fluorescence in situ hybridization (FISH)

DNA FISH for two representative chromosomes (chromosomes 2 and 16) was performed in lymphocytes, splenocytes, and primary MEFs. Cells were first incubated in hypotonic KCl solution: lymphocytes were incubated in 75 mM KCl at 37°C for 15 min, splenocytes in 75 mM KCl at room temperature for 30 min, and MEFs in 0.4% KCl at room temperature for 8 min. Subsequently, cells

were centrifuged and resuspended in methanol/acetic acid (3:1), washed twice with methanol/acetic acid (3:1), dropped onto Rite-On Micro Slides (Gold Seal Products), air-dried, and either used immediately or stored at −20°C until needed. DNA FISH was performed using the Mouse IDetect Chromosome Point Probes for chromosome 2 (red) and 16 (green) (IDMP1002-R, IDMP1016-1-G, Empire Genomics) following the manufacturer's protocol. Slides were mounted with ProLong Diamond Antifade Mountant with DAPI (Invitrogen) and analyzed on an AxioObserver Z1 microscope (Zeiss) using the 100X oil objective. Lymphocytes and splenocytes whose chromosome count differed from 2n for at least one of the two tested chromosomes were regarded as aneuploid or off mode. MEFs were only considered aneuploid when their chromosome count differed from 2n or a multiple of 2n in order to account for the increased polyploidy in this cell type.

## Time-lapse microscopy

Primary *Norad*$^{+/+}$ and *Norad*$^{-/-}$ MEFs (three MEF lines per genotype, P<4) were grown on Lab-Tek Chambered Coverglass slides (Thermo Fisher) that were coated with poly-L-lysine (Sigma-Aldrich). Prior to the analysis, DNA was visualized by adding 50 ng/mL Hoechst dye (Invitrogen) to the growth medium. Mitoses were monitored by taking fluorescence images every 5 min for ~48 hr on a Leica inverted microscope equipped with a temperature and $CO_2$-controlled chamber, a 63X oil objective, an Evolve 512 Delta EMCCD camera, and the MetaMorph Microscopy Automation and Image Analysis Software (Molecular Devices, LLC). Videos were generated from the acquired time-lapse images and analyzed for the occurrence of mitotic defects including anaphase bridges and lagging chromosomes.

## Assessment of aging-associated phenotypes

*Norad*$^{+/+}$, *Norad*$^{+/-}$, and *Norad*$^{-/-}$ mice were continuously monitored over a period of 12 months for the onset and progression of kyphosis as well as alopecia and graying of fur. The kyphosis scoring system was adopted from a previously reported strategy (*Guyenet et al., 2010*). In brief, a kyphosis score of 0 indicates no kyphosis detectable, a score of 1 indicates the presence of mild kyphosis but the mouse is still able to entirely stretch its spine, and scores of 2 and 3 indicate that there is prominent kyphosis at rest which persists in a mild (score of 2) or a severe (score of 3) form even when the mouse stretches its spine.

## Analysis of whole-body fat and subcutaneous adipose tissue

Whole-body fat, as a percentage of body weight, was measured by nuclear magnetic resonance (NMR) in 3-month and 12-month-old *Norad*$^{+/+}$ and *Norad*$^{-/-}$ mice, or in 4–6 month-old *Pum2; rtTA3* double transgenic mice and control littermates after 1.5–2 months of dox treatment, using a Bruker Minispec mq10. Subcutaneous (s.c.) adipose thickness of 12-month-old *Norad*$^{+/+}$ and *Norad*$^{-/-}$ mice was determined using standard H and E-stained skin histology. For every skin sample, images were acquired at 5X magnification across the entire length of the section on an AxioObserver Z1 microscope (Zeiss) using the AxioVision 4.8 software (Zeiss). In these images, the thickness of the s.c. adipose tissue was measured at 15 different points using the AxioVision 4.8 software. The average of the 15 measurements was then calculated to obtain the adipose thickness of each mouse.

## Quantification of mitochondrial membrane potential (MMP)

Tetramethylrhodamine ethyl ester (TMRE) (ENZ-52309, Enzo Life Sciences) was used for measuring mitochondrial membrane potential (MMP) in immortalized *Norad*$^{+/+}$ and *Norad*$^{-/-}$ MEFs. $80 \times 10^3$ cells were seeded in triplicate in 6-well plates in regular growth medium and incubated for 16–18 hr until 70–80% confluent. Cells were then trypsinized with 0.25% trypsin/EDTA (Gibco), pelleted at 300 g, and resuspended in 500 µL of fresh growth medium containing 50 nM TMRE. Samples were then incubated at 37°C in the dark for 30 min and analyzed by flow cytometry using a BD Accuri C6 Cytometer (BD Biosciences). The average and standard deviation of the mean fluorescence intensities of the three replicates was calculated for each sample, and each *Norad*$^{-/-}$ MEF line was compared to its *Norad*$^{+/+}$ littermate control line.

## Analysis of reactive oxygen species (ROS) and oxidative damage

The Enzo Total ROS Detection Kit (ENZ-51011, Enzo Life Sciences) was used for detection of ROS levels in immortalized MEFs and human HCT116 cells. $80 \times 10^3$ MEFs were plated in triplicate in -well plates in regular growth medium and incubated for 16–18 hr. $60 \times 10^3$ HCT116 cells were seeded in triplicate in 24-well plates in regular growth medium and also incubated for 16–18 hr. All cells were 70–80% confluent at the time of the assay. ROS levels were measured according to the manufacturer's protocol. Briefly, cells were trypsinized with 0.25% trypsin/EDTA (Gibco), pelleted at 300 g, washed once with Enzo 1X Wash Buffer, and resuspended in 300 μL of freshly prepared Enzo ROS Detection Solution (1 μL ROS dye in 5 mL Wash Buffer). Samples were incubated in the dark for 30 min and analyzed by flow cytometry using a BD Accuri C6 Cytometer (BD Biosciences). The average and standard deviation of the mean fluorescence intensities of the three replicates was calculated for each sample. Each $Norad^{-/-}$ MEF line was compared to its $Norad^{+/+}$ littermate control line.

Oxidative damage was examined in the CNS of 12-month-old $Norad^{+/+}$ and $Norad^{-/-}$ mice using IHC and immunofluorescence. Lipid peroxidation was examined in formalin-fixed paraffin embedded spinal cord sections by IHC using the polyclonal rabbit anti-4-Hydroxynonenal (4-HNE) antibody (ab46545, Abcam). Nucleic acid oxidation was assessed in fresh frozen brain sections by immunofluorescence using the mouse monoclonal anti-DNA/RNA Damage antibody (ab62623, Abcam), which detects 8-OHdG/8-OHG. In addition, protein oxidation was assessed in spinal cord and brain using an enzyme-linked immunosorbent assay (ELISA) for 3-nitrotyrosine (3-NT) (MBS262795, Mybiosource) according to the manufacturer's instructions.

## Seahorse analysis

Oxygen consumption rates (OCRs) were analyzed in immortalized MEFs and human HCT116 cells using a Seahorse Bioscience XF96 Extracellular Flux Analyzer (Seahorse Bioscience/Agilent). $10 \times 10^3$ cells per well were plated in Seahorse XF96 cell culture microplates (Agilent) in regular growth medium and incubated for 14–16 hr. For HCT116 cells, Seahorse XF96 Cell Culture Microplates were coated with poly-L-lysine (Sigma-Aldrich) to improve cell attachment. Prior to measurement, cells were equilibrated for 1 hr in Seahorse assay medium (D5030, Sigma-Aldrich) supplemented with 10 mM glucose (Sigma-Aldrich), 1 mM sodium pyruvate (Gibco), and 2 mM L-glutamine (Sigma-Aldrich). OCRs were monitored before and after adding the following mitochondrial inhibitors: 2 μM oligomycin (complex V inhibitor), 1 μM carbonyl cyanide 3-chlorophenylhydrazone (CCCP, uncoupler of oxidative phosphorylation), and 1 μM antimycin A (complex I II inhibitor) (all Sigma-Aldrich). OCRs were normalized to the amount of protein in each sample using a bicinchoninic acid (BCA) assay (Thermo Fisher) according to the manufacturer's instructions. For HCT116 cells, OCRs were further normalized to the mtDNA content to account for differences in mitochondrial content.

## FLAG-PUM2 overexpression in MEFs

To overexpress PUM2 in MEFs, the same $FLAG-Pum2$ cDNA that was used for generating the transgenic mouse was cloned into a pBROAD3 vector (Invivogen), in which the Rosa26 promoter was replaced by a strong CAG promoter (pCAG-$FLAG$-$Pum2$). A pcDNA3-EGFP vector was used for control transfections. FLAG-PUM2 and EGFP were transfected into immortalized $Norad^{+/+}$ MEFs using 2.5 μg of plasmid DNA and Lipofectamine 3000 (Invitrogen) according to the manufacturer's protocol. In brief, $100 \times 10^3$ cells were seeded into -well plates and incubated overnight. The next day, cells were transfected with either pCAG-$FLAG$-$Pum2$ or pcDNA3-EGFP and incubated again overnight. The following day, cells were collected and re-plated for a second transfection, performed identically. After overnight incubation, cells were seeded for Seahorse analysis, as described above. PUM2 overexpression was assessed after the second transfection using western blot.

## Statistical analysis

A comprehensive description of the RNA-seq and eCLIP analysis including the use of software is provided in the respective sections. The significance of the cumulative distribution functions was calculated using the Kolmogorov-Smirnov test and plotted in R. For all other analyses, statistical significance was analyzed using Prism 7 (GraphPad Software). Student's t tests or log-rank tests (for survival and phenotype incidence) were used to determine statistical significance, which is indicated as *$p \leq 0.05$, **$p \leq 0.01$, ***$p \leq 0.001$, ****$p \leq 0.0001$. Data are presented as mean ± SD in all figures

except *Figure 5—figure supplement 2D* and *Figure 7—figure supplement 3C* (left graph) where the data are presented as mean ± SEM. The numbers of replicates are stated in the figure legends.

## Data availability

RNA-seq and eCLIP data has been deposited in the Gene Expression Omnibus (GEO) at NCBI (Accession numbers GSE121684, GSE121688, and GSE125539). Data is available for download via the following links:

https://www.ncbi.nlm.nih.gov/geo/query/acc.cgi?acc=GSE121684
https://www.ncbi.nlm.nih.gov/geo/query/acc.cgi?acc=GSE121688
https://www.ncbi.nlm.nih.gov/geo/query/acc.cgi?acc=GSE125539

## Acknowledgements

We thank Feng Zhang for plasmids, Shinichi Nakagawa for technical assistance with RNA FISH, Jeanetta Marshburn-Wynn for assistance with mouse husbandry, Robert Hammer and the UT Southwestern Transgenic Core for assistance with mouse generation, James Richardson, John Shelton, Cheryl Lewis, the UT Southwestern Histopathology Core, and the UT Southwestern Tissue Management Shared Resource for assistance with histopathology, Vanessa Schmid and the McDermott Center Next Generation Sequencing Core for high-throughput sequencing, Orhan Oz and Xiankai Sun from the UT Southwestern Department of Radiology for mouse imaging, the Experimental Neuromuscular Laboratories in the Center for Gene Therapy at Nationwide Children's Hospital for technical support, Jose Cabrera for assistance with graphics, and Kathryn O'Donnell, Eric Olson, and members of the Mendell laboratory for helpful comments on the manuscript. This work was supported by grants from CPRIT (RP160249 to JTM; RP150596 for the UTSW Bioinformatics Core Facility), NIH (R35CA197311 to JTM; P30CA142543 to JTM and the UT Southwestern Tissue Resource; and P50CA196516 to JTM), and the Welch Foundation (I-1961–20180324 to JTM). FK is supported by the Leopoldina Fellowship Program (LPDS 2014–12) from the German National Academy of Sciences Leopoldina. JTM and HY are Investigators of the Howard Hughes Medical Institute.

## Additional information

### Funding

| Funder | Grant reference number | Author |
| --- | --- | --- |
| Howard Hughes Medical Institute | | Hongtao Yu<br>Joshua T Mendell |
| National Institutes of Health | R35CA197311 | Joshua T Mendell |
| Cancer Prevention and Research Institute of Texas | RP160249 | Yang Xie<br>Joshua T Mendell |
| Welch Foundation | I-1961-20180324 | Joshua T Mendell |
| German National Academy of Sciences Leopoldina | LPDS 2014-12 | Florian Kopp |
| National Institutes of Health | P30CA142543 | Joshua T Mendell |
| National Institutes of Health | P50CA196516 | Joshua T Mendell |
| Cancer Prevention and Research Institute of Texas | RP150596 | Yang Xie<br>Joshua T Mendell |

The funders had no role in study design, data collection and interpretation, or the decision to submit the work for publication.

### Author contributions

Florian Kopp, Conceptualization, Data curation, Formal analysis, Funding acquisition, Validation, Investigation, Visualization, Methodology, Writing—original draft, Project administration, Writing—review and editing; Mahmoud M Elguindy, Formal analysis, Investigation, Methodology; Mehmet E Yalvac, Sushama Sivakumar, Data curation, Formal analysis, Investigation, Visualization,

Methodology; He Zhang, Beibei Chen, Data curation, Software, Formal analysis, Visualization, Methodology; Frank A Gillett, Validation, Investigation, Visualization; Sungyul Lee, Conceptualization, Data curation, Formal analysis, Validation, Investigation, Methodology; Hongtao Yu, Conceptualization, Resources, Supervision, Methodology; Yang Xie, Conceptualization, Resources, Supervision; Prashant Mishra, Conceptualization, Resources, Data curation, Formal analysis, Investigation, Methodology; Zarife Sahenk, Conceptualization, Resources, Data curation, Formal analysis, Supervision, Investigation, Visualization, Methodology, Writing—review and editing; Joshua T Mendell, Conceptualization, Resources, Supervision, Funding acquisition, Investigation, Methodology, Writing—original draft, Project administration, Writing—review and editing

## Author ORCIDs
Florian Kopp (iD) http://orcid.org/0000-0001-9952-635X
Mahmoud M Elguindy (iD) https://orcid.org/0000-0001-9151-1751
Sungyul Lee (iD) http://orcid.org/0000-0003-3207-1199
Sushama Sivakumar (iD) http://orcid.org/0000-0001-7877-4821
Hongtao Yu (iD) http://orcid.org/0000-0002-8861-049X
Joshua T Mendell (iD) http://orcid.org/0000-0001-8479-2284

## Ethics
Animal experimentation: This study was performed in strict accordance with the recommendations in the Guide for the Care and Use of Laboratory Animals of the National Institutes of Health. All animals were handled according to approved institutional animal care and use committee (IACUC) protocols of The University of Texas Southwestern Medical Center (Animal Protocol Number 2017-102001) and The Ohio State University, Nationwide Children's Hospital (Animal Protocol Number AR12-00014).

## Decision letter and Author response
Decision letter https://doi.org/10.7554/eLife.42650.032
Author response https://doi.org/10.7554/eLife.42650.033

# Additional files
## Supplementary files
• Supplementary file 1. PUM2 eCLIP clusters in *Norad*<sup>+/+</sup> and *Norad*<sup>−/−</sup> brain. Genomic coordinates refer to mm10.
DOI: https://doi.org/10.7554/eLife.42650.020
• Supplementary file 2. Sequences of oligonucleotides used in this study.
DOI: https://doi.org/10.7554/eLife.42650.021
• Transparent reporting form
DOI: https://doi.org/10.7554/eLife.42650.022

## Data availability
RNA-seq and eCLIP data has been deposited in the Gene Expression Omnibus (GEO) at NCBI (Accession numbers GSE121684, GSE121688, and GSE125539).

The following datasets were generated:

| Author(s) | Year | Dataset title | Dataset URL | Database and Identifier |
| --- | --- | --- | --- | --- |
| Kopp F, Chen B, Zhang H, Lee S, Xie Y, Mendell JT | 2018 | Identification of RNAs bound to PUM2 in Norad+/+ and Norad-/- brains [CLIP-seq] | https://www.ncbi.nlm.nih.gov/geo/query/acc.cgi?acc=GSE121684 | NCBI Gene Expression Omnibus, GSE121684 |
| Kopp F, Chen B, Zhang H, Lee S, Xie Y, Mendell JT | 2018 | Gene expression profiles in Norad +/+ and Norad-/- brains and spleens [RNA-seq] | https://www.ncbi.nlm.nih.gov/geo/query/acc.cgi?acc=GSE121688 | NCBI Gene Expression Omnibus, GSE121688 |
| Kopp F, Chen B, | 2019 | Gene expression profiles in double | https://www.ncbi.nlm. | NCBI Gene |

| Zhang H, Lee S | | transgenic (DT, Pum2;rtTA3) and control (CTR, Pum2 and wild-type) spleens | nih.gov/geo/query/acc. cgi?acc=GSE125539 | Expression Omnibus, GSE125539 |
|---|---|---|---|---|

The following previously published dataset was used:

| Author(s) | Year | Dataset title | Dataset URL | Database and Identifier |
|---|---|---|---|---|
| Kopp F, Chang T, Chen B, Xie Y, Mendell JT | 2015 | Gene expression profiles in NORAD knockout and PUMILIO overexpressing cells | https://www.ncbi.nlm. nih.gov/geo/query/acc. cgi?acc=GSE75440 | NCBI Gene Expression Omnibus, GSE75440 |

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
