## [Decision Letter]

Thank you for submitting your article "PUMILIO hyperactivity drives premature aging of *Norad*-deficient mice" for consideration by *eLife*. Your article has been reviewed by three peer reviewers, including Chris P Ponting as the Reviewing Editor and Reviewer #1, and the evaluation has been overseen by James Manley as the Senior Editor. The following individuals involved in review of your submission have agreed to reveal their identity: Igor Ulitsky (Reviewer #2). A further reviewer remains anonymous.

The reviewers have discussed the reviews with one another and the Reviewing Editor has drafted this decision to help you prepare a revised submission.

In this manuscript, Mendell and colleagues describe the generation and characterization of Norad-/- mice. NORAD was previously described as an abundant and conserved lncRNA, whose disruption in human cells results in genome instability through either increase in activity of Pumilio proteins in the cytoplasm or disruption of protein complex formation involving RBMX in the nucleus. In this new work, Norad-/- mice were generated, and shown to have a severe aging phenotype, which is accompanied by genome instability and mitochondrial defects, along with some changes in gene expression, which can be attributed in part to hyperactivity of the Pumilio proteins. Convincingly, mice with mild inducible transgenic over-expression of Pum2 are shown to recapitulate the Norad-/- phenotype, strongly arguing that Pumilio hyperreactivity is indeed responsible for the Norad loss-of-function phenotype.

Essential Revisions:

1) CLIP experiments.

A) Arguably the weakest part of the results is the changes observed in gene expression in the mouse brain, particularly Figure 3E, which shows deviations in both up-regulated and down-regulated genes, contrary to the expectation for mostly up-regulation of Pumilio targets. This is possibly because the authors select Pum targets based on CLIP data, which has the advantage of yielding "experimentally defined" targets, but biases towards more abundantly expressed genes (as those are likely to have more reads/clusters in CLIP data). What does Figure 3E look like if the authors define the targets based on presence of PREs in 3'UTRs (e.g., requiring at least 2 PREs)? An alternative would be to sample non-targets that have the same expression level distribution (and 3' UTR lengths) as the CLIP-defined targets and repeat the analysis. In any case, it is worth discussing how the changes observed in gene expression, at least in bulk tissue are very small (~5%) and occur in both directions. This is consistent with the studies in human cells, but still worth dwelling on in the Discussion section.

B) Another molecular phenotype the authors observe is increase in Pumilio target clusters based on eCLIP data. This analysis has two caveats – first, differences in quality of eCLIP libraries can result in differences in numbers of clusters etc. Can the authors show that the replicates of eCLIP from the same genotype (if performed) are similar to each other more than Norad+/+ samples to Norad-/- samples? Alternatively – is the PRE motif recovered equally strongly from Norad+/+ and Norad-/- libraries?

C) The other caveat with the eCLIP analysis is that the authors normalize the total number of eCLIP reads, and then compare the eCLIP/FPKM ratios between genes. This makes it difficult to interpret Figure 3D. To me, it shows that the reads become redistributed to clusters in the more abundant genes, rather than that target occupancy globally is indeed increased (it’s not clear to me that the latter can be shown with just eCLIP data, which have to be normalized somehow). First, why don't both curves reach 1 at the y-axis? Also, since the total number of reads is normalized, is this evidence sufficient to say that "target occupancy was significantly increased"? The authors should explain more thoroughly why this analysis is valid (I could be missing something), and how it supports the conclusion in the main text.

D) Related to (A) above. Is a PRE-dependent signature of Pum target repression evident in the MEF data (Figure 4)? If it is – it's worth showing in supplement. If not – worth discussing.

2) NORAD/RBMX.

Analysis of two other protein binding partners have been published for Norad/NORAD: SAM68 and RBMX. In particular, the interaction of NORAD with RBMX and its influence on the topoisomerase complex (Munschauer et al., 2018) is very convincing and accounts for the genome instability resulting from NORAD loss. Moreover, the topoisomerase complex is also implicated in mtDNA maintenance, similar to what the authors show in their work here. The authors briefly mention the NORAD/RBMX study in the Introduction, but curiously state that the role of this interaction for genome stability is unknown. The role of topoisomerases in genome stability is thus known and the authors need to discuss the NORAD/RBMX study in light of their own findings. There are, however, some discrepancies between these studies. Lee et al., and Tichon et al., found NORAD to be predominantly localized in the cytoplasm, while the NORAD/RBMX study find that NORAD is evenly distributed (smFISH and fractionation) between the nucleus and the cytoplasm. RNA-FISH data, if available, would resolve this issue.

3) PUM2 level in FLAG-tag experiment.

There is a surprising lack of PUM2 protein level change for the FLAG-tag PUM2 mouse experiment. This means that the authors' model is not definitive and they need to be more circumspect in their conclusions. (subsection “Enforced PUM2 expression phenocopies *Norad* loss of function”) The authors need to define how PUM2 is deregulated, if it is not overtly overexpressed. Otherwise, the models cannot be defined as such. One alternative hypothesis is that deregulated PUM2 RNA, but not protein, is required for the phenotype to be manifested. Do the authors have evidence that this is, or is not, the case? If not then they need to discuss alternative explanations, remove the word "accordingly" (Abstract), and insert further caveats to their conclusion that (subsection “Enforced PUM2 expression phenocopies *Norad* loss of function” and elsewhere) "PUMILIO hyperactivity in Norad-deficient animals results in.…" etc. The rare cell specificity of effect (Discussion section) is an attractive one but is not supported by evidence in the submission. Subsection “Enforced PUM2 expression phenocopies *Norad* loss of function”: The lack of alteration of PUM2 protein levels in these 7 tissues (even with the negative feedback mechanism) cannot be reconciled with the PUMILIO "hyperactivity" of the title.

---

## [Author Response]

Essential Revisions:1) CLIP experiments.A) Arguably the weakest part of the results is the changes observed in gene expression in the mouse brain, particularly Figure 3E, which shows deviations in both up-regulated and down-regulated genes, contrary to the expectation for mostly up-regulation of Pumilio targets. This is possibly because the authors select Pum targets based on CLIP data, which has the advantage of yielding "experimentally defined" targets, but biases towards more abundantly expressed genes (as those are likely to have more reads/clusters in CLIP data). What does Figure 3E look like if the authors define the targets based on presence of PREs in 3'UTRs (e.g., requiring at least 2 PREs)? An alternative would be to sample non-targets that have the same expression level distribution (and 3' UTR lengths) as the CLIP-defined targets and repeat the analysis. In any case, it is worth discussing how the changes observed in gene expression, at least in bulk tissue are very small (~5%) and occur in both directions. This is consistent with the studies in human cells, but still worth dwelling on in the Discussion section.

We appreciate these excellent suggestions, which have greatly improved the gene expression analyses in the paper. As noted by the reviewers, our original analysis of expression of PUM2 CLIP targets vs. non-targets in *Norad^+/+^* and *Norad^–/–^* brains simply compared all CLIP targets to all non-targets. To incorporate the reviewers’ suggestions, we first filtered the CLIP targets for those with at least 2 PREs in the 3’ UTR. However, this only marginally increased the apparent repression of PUM2 targets in *Norad^–/–^* brain (not shown). On the other hand, the reviewers’ suggestion to sample non-targets with the same expression level distribution and 3’ UTR length dramatically improved the results. As shown in the new Figure 3D-E, when non-targets are filtered for those whose expression level and 3’ UTR length are similar to these parameters in the set of PUM2 CLIP targets, the results much more clearly show repression of PUM2 targets in *Norad^–/–^* brain. In addition, we have expanded the discussion in the manuscript to introduce the concept that modest repression of a large set of PUMILIO targets can, in aggregate, produce a dramatic phenotype.

B) Another molecular phenotype the authors observe is increase in Pumilio target clusters based on eCLIP data. This analysis has two caveats – first, differences in quality of eCLIP libraries can result in differences in numbers of clusters etc. Can the authors show that the replicates of eCLIP from the same genotype (if performed) are similar to each other more than Norad+/+ samples to Norad-/- samples? Alternatively – is the PRE motif recovered equally strongly from Norad+/+ and Norad-/- libraries?

We agree that comparing the absolute numbers of PUMILIO CLIP clusters between *Norad^+/+^* and *Norad^–/–^* samples could be easily confounded by experimental variation. Therefore, we did not intend to make the point that there were significantly more clusters in *Norad^–/–^* brain. We did not put forth this conclusion in the original text, although the original Figure 3C which showed the number of CLIP clusters in each genotype may have inadvertently implied that we were interpreting any apparent differences as meaningful. To avoid any confusion, we have removed this figure panel from the revised manuscript. The numbers of CLIP clusters in each genotype can be easily determined from Supplementary file 1 should a reader be interested in this information.

C) The other caveat with the eCLIP analysis is that the authors normalize the total number of eCLIP reads, and then compare the eCLIP/FPKM ratios between genes. This makes it difficult to interpret Figure 3D. To me, it shows that the reads become redistributed to clusters in the more abundant genes, rather than that target occupancy globally is indeed increased (it’s not clear to me that the latter can be shown with just eCLIP data, which have to be normalized somehow). First, why don't both curves reach 1 at the y-axis? Also, since the total number of reads is normalized, is this evidence sufficient to say that "target occupancy was significantly increased"? The authors should explain more thoroughly why this analysis is valid (I could be missing something), and how it supports the conclusion in the main text.

We apologize for any confusion regarding this analysis. To generate this figure, we first normalized the number of CLIP reads in each library based on the total number of usable reads (uniquely mapped, non-PCR duplicates). We next calculated the average number of normalized CLIP reads in each 3’ UTR cluster between the two replicates that were performed for each genotype. For each PUM2 CLIP target, we determined the total number of clusters in the 3’ UTR and then summed the average number of normalized CLIP reads in each cluster to determine the total number of normalized CLIP reads per CLIP target. Because increased PUM2 binding (expected to occur frequently in the *Norad^–/–^* condition) will likely result in reduced steady-state mRNA abundance, it is critical to normalize the apparent CLIP signal to the expression of the transcript. Therefore, we divided the normalized CLIP reads per target by FPKM (note that we are only considering targets expressed at or above 1 FPKM). These data were then used to generate the CDF plot in Figure 3C (previously Figure 3D). Importantly, this method of analysis has been used previously to estimate relative binding in CLIP data (Bosson et al., 2014) and we believe that it can provide a reasonable estimate of this parameter. To be as careful as possible, we have expanded our discussion of this result in the text to better describe how the analysis was done and to emphasize that it provided an estimate of relative CLIP signal that suggested increased PUM2 target occupancy in *Norad^–/–^* brain. We have also added the Bosson et al., reference to the main text and Materials and methods section.

Regarding the question of why both curves do not reach 1 at the y-axis in Figure 3C of the current manuscript, we simply zoomed in on a narrower region of the x-axis to better highlight the difference between *Norad^+/+^* and *Norad^–/–^* conditions, as is commonly done with these types of plots. The full plot is provided here (Author response image 1) for the reviewers.

D) Related to (A) above. Is a PRE-dependent signature of Pum target repression evident in the MEF data (Figure 4)? If it is – it's worth showing in supplement. If not – worth discussing.

To clarify, the gene expression data in Figure 4 is derived from RNA-seq of *Norad^+/+^* and *Norad^–/–^* spleens (a representative mitotic tissue which shows genomic instability in knockout animals). Regardless, whether there is evidence of enhanced PUM2 target repression in these data is a valid question which we have addressed in the revised manuscript. Starting with the set of PUM2 brain CLIP targets identified in this study, we first determined which of these were expressed in spleen (FPKM ≥ 1). Because we were attempting to translate the targets identified in brain to a different tissue, we further filtered the CLIP targets for those that have at least two PREs (building off of a suggestion from the reviewers above). The expression of this set of PUM2 targets was compared to the set of non-targets with similar expression level and 3’ UTR length. The results, presented in Figure 4E of the revised manuscript, demonstrate statistically-significant repression of PUM2 targets in *Norad^–/–^* spleen.

2) NORAD/RBMX.Analysis of two other protein binding partners have been published for Norad/NORAD: SAM68 and RBMX. In particular, the interaction of NORAD with RBMX and its influence on the topoisomerase complex (Munschauer et al., 2018) is very convincing and accounts for the genome instability resulting from NORAD loss. Moreover, the topoisomerase complex is also implicated in mtDNA maintenance, similar to what the authors show in their work here. The authors briefly mention the NORAD/RBMX study in the Introduction, but curiously state that the role of this interaction for genome stability is unknown. The role of topoisomerases in genome stability is thus known and the authors need to discuss the NORAD/RBMX study in light of their own findings. There are, however, some discrepancies between these studies. Lee et al., and Tichon et al., found NORAD to be predominantly localized in the cytoplasm, while the NORAD/RBMX study find that NORAD is evenly distributed (smFISH and fractionation) between the nucleus and the cytoplasm. RNA-FISH data, if available, would resolve this issue.

We agree that the *NORAD*:RBMX interaction is very interesting and we did not mean to diminish those findings. It is important to point out that the regulation of RBMX and PUMILIO by *NORAD* are not mutually exclusive and it is possible that this lncRNA performs both functions. We have therefore revised this section of the Introduction by describing the role of SAM68 in PUMILIO antagonism by *NORAD* and by emphasizing that the intriguing RBMX interaction suggests other functions for *NORAD* beyond regulating PUMILIO activity.

Despite these revisions, we still feel that it is appropriate and important to state that it has not yet been demonstrated that the *NORAD*:RBMX interaction is essential to maintain genomic stability. The basis for this statement is the fact that no experimental evidence exists showing that loss of the RBMX binding site in *NORAD* impairs *NORAD*’s ability to maintain genome stability. Alternatively, one could show a critical role for RBMX in the *NORAD* loss-of-function phenotype by performing genetic epistasis experiments, in which modulation of RBMX activity is shown to rescue the *NORAD* deletion phenotype. These types of experiments have not yet been performed. It is important to contrast this with the *NORAD*:PUMILIO interaction, for which it has been shown that PUMILIO depletion reverses the *NORAD* loss-of-function phenotype (Lee et al., 2016), providing strong genetic evidence for PUMILIO’s role in the *NORAD* pathway. Our purpose in making this statement is to provide as accurate a description of the state of the field as possible. Nevertheless, given that our findings reported here do not in any way address the role of RBMX in the *NORAD* pathway, we do not think it is appropriate to further address that work in this manuscript.

In addition, as suggested by the reviewers, we have performed further experiments to investigate the localization of *Norad* in mouse cells. RNA FISH in MEFs clearly shows a predominantly cytoplasmic localization (new Figure 1C), in agreement with the localization of the human transcript reported in Lee et al. and Tichon et al. We also performed additional fractionation experiments in a total of four mouse cell lines (new Figure 1—figure supplement 1C), which yielded results that perfectly agree with our RNA FISH data and show ~80% cytoplasmic localization of *Norad* (equivalent to *Actb* mRNA) in all cell lines tested.

3) PUM2 level in FLAG-tag experiment.There is a surprising lack of PUM2 protein level change for the FLAG-tag PUM2 mouse experiment. This means that the authors' model is not definitive and they need to be more circumspect in their conclusions. (subsection “Enforced PUM2 expression phenocopies Norad loss of function”) The authors need to define how PUM2 is deregulated, if it is not overtly overexpressed. Otherwise, the models cannot be defined as such. One alternative hypothesis is that deregulated PUM2 RNA, but not protein, is required for the phenotype to be manifested. Do the authors have evidence that this is, or is not, the case? If not then they need to discuss alternative explanations, remove the word "accordingly" (Abstract), and insert further caveats to their conclusion that (subsection “Enforced PUM2 expression phenocopies Norad loss of function” and elsewhere) "PUMILIO hyperactivity in Norad-deficient animals results in.…" etc. The rare cell specificity of effect (Discussion section) is an attractive one but is not supported by evidence in the submission. Subsection “Enforced PUM2 expression phenocopies Norad loss of function”: The lack of alteration of PUM2 protein levels in these 7 tissues (even with the negative feedback mechanism) cannot be reconciled with the PUMILIO "hyperactivity" of the title.

We have now performed a number of new experiments that have further documented the expression and activity of FLAG-PUM2 in our transgenic mouse lines and provide additional evidence that these animals indeed represent a model of PUMILIO hyperactivity. First, as noted by the reviewers, we posited that the inability to detect an increase in PUM2 levels in bulk tissue in transgenic mice may be due to the masking of transgene induction in specific key cell types due to the complex mixture of cell types in a tissue. To address this model, we isolated MEF lines from transgenic animals to interrogate whether transgene expression is detectable in a more uniform cell population. These results, provided in the new Figure 7—figure supplement 1B, clearly demonstrate robust induction of FLAG-PUM2 protein and a corresponding increase in total PUM2 levels in *Pum2; rtTA3* MEFs from both transgenic lines. Thus, transgene activation leads to PUM2 overexpression in specific cell types. To demonstrate that transgene activation leads to PUM2 hyperactivity in vivo, we went a step further by performing RNA-seq in spleens of *Pum2; rtTA3* animals. This revealed robust repression of PUM2 targets in mice with transgene activation (new Figure 7—figure supplement 1D). These results provide clear evidence of PUM2 hyperactivity in dox-treated *Pum2; rtTA3* mice.

In addition to the new data demonstrating PUM2 hyperactivity in cells and tissues from transgenic mice, we would also like to emphasize that these in vivo studies are complemented by our data derived from overexpressing PUM1 and PUM2 in human and mouse cell lines (new Figure 7; Figure 7—figure supplement 2; Figure 7—figure supplement 3). These studies show that overexpression of PUM1 or PUM2 protein impairs mitochondrial function, providing further evidence to support the conclusion that PUMILIO hyperactivity drives the observed phenotypes. These new data are also consistent with our previously reported finding that overexpression of PUM2 protein is sufficient to drive genomic instability in human cell lines (Lee et al., 2016). On the basis of this body of evidence, we feel that it is justifiable to refer to “PUMILIO hyperactivity” in the title.